# PROOFFLOW: A DEPENDENCY GRAPH APPROACH TO FAITHFUL PROOF AUTOFORMALIZATION

**Rafael Cabral**[1]**, Tuan Manh Do**[1]**, Xuejun Yu**[1]**, Wai Ming Tai**[1]**, Zijin Feng**[2]**, Xin Shen**[1,*]
[1]Huawei Celia Team
[2]Huawei Foundation Model Department
shenxin19@huawei.com

## ABSTRACT

Proof autoformalization, the task of translating natural language theorems and proofs into machine-verifiable code, is a critical step for integrating large language models into rigorous mathematical workflows. Current approaches focus on producing executable code, but they frequently fail to preserve the semantic meaning and logical structure of the original human-written argument. To address this, we introduce PROOFFLOW, a novel pipeline that treats structural fidelity as a primary objective. PROOFFLOW first constructs a directed acyclic graph (DAG) to map the logical dependencies between proof steps. Then, it employs a novel lemma-based approach to systematically formalize each step as an intermediate lemma, preserving the logical structure of the original argument. To facilitate evaluation, we present a new benchmark of 184 undergraduate-level problems, manually annotated with step-by-step solutions and logical dependency graphs, and introduce PROOFSCORE, a new composite metric to evaluate syntactic correctness, semantic faithfulness, and structural fidelity. Experimental results show our pipeline sets a new state-of-the-art for autoformalization, achieving a PROOFSCORE of 0.545, substantially exceeding baselines like full-proof formalization (0.279), which processes the entire proof at once, and step-proof formalization (0.046), which handles each step independently. Our pipeline, benchmark, and score metric are open-sourced to encourage further progress at https://github.com/Huawei-AI4Math/ProofFlow.

## 1 INTRODUCTION

The effort to automate mathematical reasoning is advancing on two key fronts. Recent advances in large language models (LLMs) have greatly enhanced their ability to solve mathematical problems (Liang et al., 2025). Meanwhile, symbolic engines such as Lean (Moura & Ullrich, 2021) and Isabelle (Hales et al., 2017) provide machine-verifiable frameworks that enforce strict logical correctness. LLM-based automated theorem provers (ATPs), such as Goedel-Prover (Lin et al., 2025) and Kimina-Prover (Wang et al., 2025), generate formal proofs for problems that are written in a symbolic language, which are then checked by the symbolic engine to ensure logical correctness.

This paper studies automated proof formalization: the task of faithfully translating the natural language theorem and proof into a machine-verifiable formal representation. This task is distinct from the aforementioned ATPs, where the goal is to discover a proof from an already formalized problem. Here, the goal is to translate an existing proof, a crucial step for verification in real-world mathematical workflows. After composing a proof, a mathematician may wish to formalize it to verify its correctness, uncover missing assumptions, or fill logical gaps. Given the steep learning curve of formal languages like Lean, a system that can automate this translation is highly desirable. The manual effort these automated proof formalizers seek to automate is immense, as demonstrated by landmark efforts such as the Flyspeck project, which took over 20 years to formalize the Kepler conjecture (Hales et al., 2017); the Blue-Diamond project for the Polynomial Freiman-Ruzsa conjecture (Tao, 2023); and the ongoing formalization of Fermat's Last Theorem (Buzzard & Taylor, 2025), among other projects (Math Inc., 2025; Scholze, 2021; Gonthier et al., 2013).

---

∗ Corresponding author.

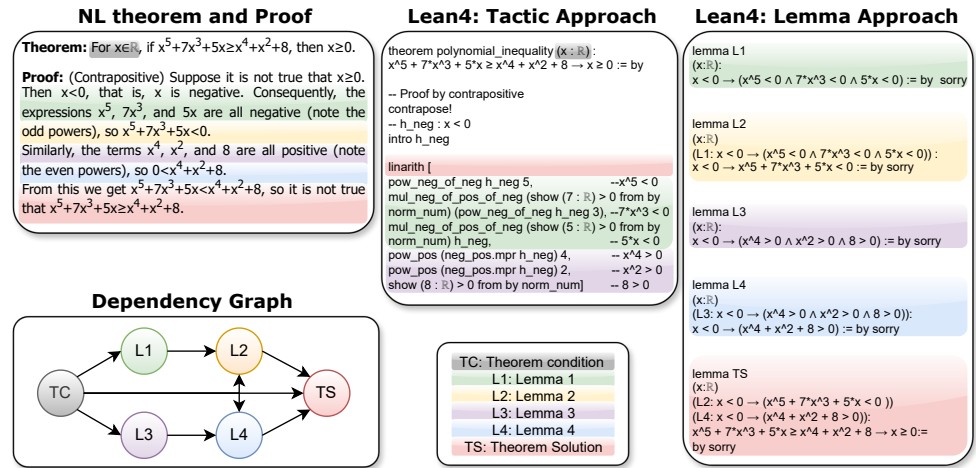

Figure 1: Comparison of our PROOFFLOW "Lemma Approach" and the common "Tactic Approach" in formalizing a natural language (NL) theorem and its proof into Lean 4 code. The Lemma Approach directly mirrors the sequence of steps and their dependencies in the NL proof. By contrast, the Tactic Approach produces tactics that fail to adhere to the structure of the initial NL proof.

A key challenge in autoformalization is the fundamental mismatch between the flexible and high-level nature of human language and the rigid, low-level syntax of formal systems. To bridge this gap, ATPs typically generate "tactics" for formal proof assistants like Lean 4. However, this tactic-based approach often fails to preserve the logical structure of the original human argument. For instance, as shown in Figure 1, under "Lean 4: Tactic Approach", the generated tactics do not follow the sequence of steps in the natural language proof. Furthermore, a single tactic, *linarith*, consolidates three distinct steps from the original proof (TS, L2, and L4). This disparity creates two major issues. First, not all natural language mathematical expressions can be directly translated into low-level tactics, which can cause the formalization process to fail. Second, even if an automated system generates a verifiable proof, it may take shortcuts or skip intermediate steps, arriving at the correct conclusion without mirroring the explicit, step-by-step reasoning of the original proof. This makes it difficult to verify that the formal proof truly captures the human's intended logic.

To address these issues and ensure faithful autoformalization, we propose a novel approach that avoids direct translation into limited, low-level formal tactics. Instead, we deconstruct the natural language proof into a sequence of structured, high-level lemmas, as illustrated by the "Lemma Approach" in Fig. 1. The key advantage of this method is that it enables us to explicitly follow the sequence of steps in the original proof with low friction[1]. This stands in sharp contrast to the conventional tactic-based approach, which is often convoluted and hinders mirroring the human proof structure. Second, our lemma-based approach preserves the logical structure of the natural language proof by explicitly defining dependencies. As shown in Figure 1, the final step, lemma TS (the theorem's solution), depends only on lemma L2 and lemma L4. This explicit dependency management is crucial in enforcing structurally faithful formalization, which is often not achieved in a standard tactic-based workflow. For instance, a common tactic like "have" in Lean 4 feeds the entire preceding context to each new step, not just the specific premises required by the original proof logic. In contrast, our approach of explicitly defining how lemmas depend on one another prevents the system from incorrectly using unintended dependencies, a common failure in tactic-based proving. These two innovations, high-level lemma generation and explicit dependency tracking, form the core of our new autoformalization pipeline, PROOFFLOW.

Another primary challenge is defining what constitutes a "faithful" formalization. In existing work, researchers either focus on syntactic correctness (Hu et al., 2025), i.e., no compilation errors, or

---

[1]While we currently use the placeholder "by sorry" for unproven parts, our ultimate goal is to generate Lean tactics to prove each lemma and the final solution.

use a simple BLEU score for semantic measurement (Poiroux et al., 2024; Wu et al., 2022), while ignoring the structural fidelity of the proof. To properly evaluate proof formalizations, we propose viewing a proof not as a monolithic block of text, but as a structured sequence of theorem conditions, definitions, and lemmas that form a logical progression toward the final theorem solution or solutions (see Figure 1). Based on this, we introduce a new and more comprehensive proof autoformalization scoring metric (PROOFSCORE), and we also address the lack of an advanced benchmark by providing a new university-level dataset tailored for this task (PROOFFLOWBENCH).

Drawing on our new framework, we make the following key contributions:

- PROOFFLOW: In Section 3, we propose a novel pipeline for translating natural language theorems and proofs into structured and formal Lean code, ensuring the preservation of the proof's logical structure. When a formalization step fails, the pipeline identifies the error source, be it in the formalization, the tactic completion process, or the initial NL statement, thereby alerting mathematicians of potential flaws in their original proof.

- PROOFSCORE: Section 4 introduces a new and comprehensive scoring method to evaluate the quality of autoformalized proofs. This metric is the first to explicitly measure syntactic correctness, semantic faithfulness, and structural fidelity, providing a more complete assessment than existing methods.

- PROOFFLOWBENCH: In Section 5, we present a new, manually curated benchmark dataset for proof autoformalization, containing a collection of 184 undergraduate level problems.

- Comparative Study: Section 6 presents an empirical study using state-of-the-art models to evaluate our structure-aware pipeline, PROOFFLOW, against alternative strategies. The results show that PROOFFLOW has significantly higher proof autoformalization quality.

## 2 BACKGROUND AND RELATED WORK

**Proof assistants and automatic theorem proving:** Proof assistants like Isabelle (Paulson, 1994), Lean 4 (Moura & Ullrich, 2021), and Coq (Barras et al., 1997) are software environments for developing and verifying mathematical proofs. Proofs constructed within these systems are what we call "formal" proofs, distinguishing them from informal or natural language proofs written in standard mathematics (e.g., in LATEX). The user's workflow involves interactively applying tactics, which are small programs that perform logical inferences, to solve the theorem's goals (Jiang et al., 2024). Despite their power, a steep learning curve and the significant manual effort required by their rigid syntaxes have hindered widespread adoption (Zhou et al., 2024). Recently, Large Language Models (LLMs) have emerged as powerful automated theorem provers (ATPs) capable of generating complete formal proofs from already formalized theorem statements (Shang et al., 2025; Lin et al., 2025; Wang et al., 2025; Ren et al., 2025). Proof Agents have further advanced this ability (Chen et al., 2025; Zhou et al., 2025; Baba et al., 2025). In the context of Figure 1, the task of an ATP is to automatically replace the placeholder command "sorry" with formal tactics to complete the proof.

**Autoformalization with LLMs:** For a fully automatic theorem-proving system, mathematical problems originating in natural language must first be translated into a formal language, a process known as autoformalization. Historically, autoformalization efforts have primarily focused on translating theorem statements, and not the natural language proof (Huang et al., 2025; Wu et al., 2025; Liu et al., 2025; Yu et al., 2025b; Poiroux et al., 2024; Pathak, 2024; Patel et al., 2023), often to support the training of automated provers (Lin et al., 2025; Wang et al., 2025). A different approach uses informal proof sketches to guide an LLM's search for a formal proof (Cao et al., 2025; Zhou et al., 2024; Jiang et al., 2023). In these methods, the natural language proof sketch is not the target of formalization itself. Rather, it serves as a high-level guide, often interleaved as comments within the formal code to steer the generation process. This guidance technique is also employed by recent ATPs like DeepseekProver-V2 (Ren et al., 2025) and Goedel Prover V2 (Lin et al., 2025).

**Proof Autoformalization:** In the literature of proof autoformalization, most existing attempts directly translate entire proofs using LLMs (Gao et al., 2024; Lu et al., 2024; Cunningham et al., 2023). This approach, however, remains highly challenging due to frequent syntactic errors and it often produces outputs that are not semantically trustworthy. Step-level formalization, which involves solving the autoformalization step-by-step based on the proof's logical steps, has been explored by Hu et al.

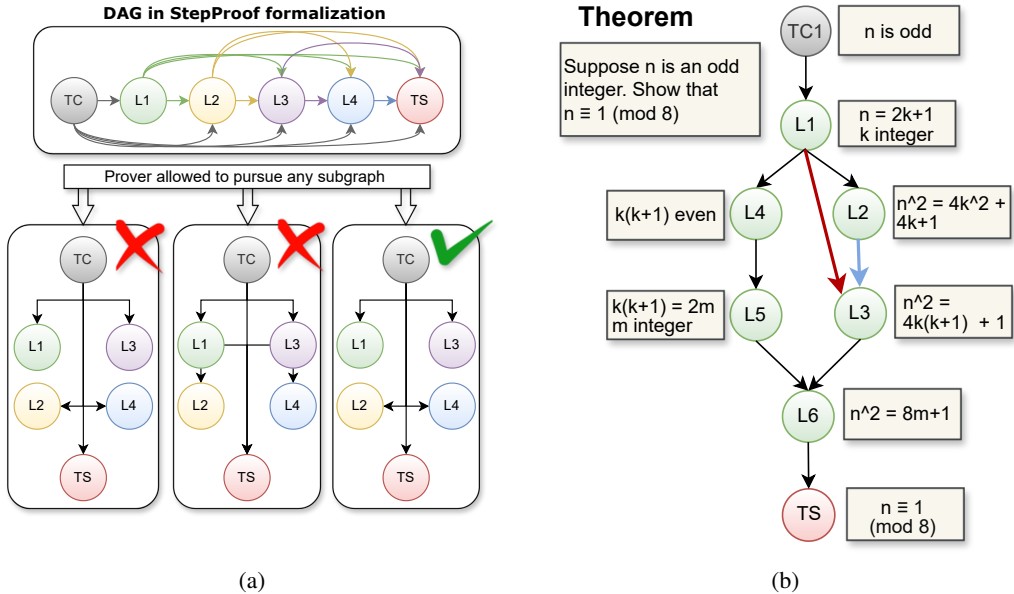

Figure 2: Comparing structural fidelity in automated proof generation. (a) A scenario for the problem in Figure 1, where the dependency graph fails to adhere to the structure of the original proof. (b) This problem was intentionally formalized without enforcing a DAG. This approach sacrificed structural fidelity, by reusing lemma L1 to prove L3, thereby rendering L2 redundant.

(2025) in Isabelle. However, this approach suffers from two limitations. First, they treated all previous proof steps as valid premises for the current step, a significant simplification that overlooks the proper logical structure. Second, their evaluations focused solely on syntactic correctness while overlooking semantic consistency.

## 3    PROOF AUTOFORMALIZATION

Although most prior work formalizes entire proofs at once, a notable exception is the "STEP PROOF" method (Hu et al., 2025). This approach assumes each proof step depends on all preceding steps, a simplification that creates an unfaithful dependency structure and can lead to an ATP taking a "shortcut." This shortcut may involve using only the initial theorem conditions or an incorrect subset of lemmas to construct a valid proof that does not follow the logic of the original NL proof, as depicted in Figure 2a. A clear example found in PROOFFLOWBENCH is illustrated in Figure 2b. In the natural language proof, step L3 directly uses the outcome of L2, which is faithfully followed by our DAG-enforcing pipeline. By contrast, an ablated version of our pipeline which lacks the mechanism to enforce the correct dependencies (noDAG) reuses the outcome of L1 to prove L3, despite having proven L2. Thus, it not only dissipates step L2 but also disregards the structure of the original proof. More details and examples are found in Appendix A.5.

If the sole objective is to find any correct formal proof for a theorem, which is often the sole goal of ATPs, these behaviors are not problematic. However, for proof autoformalization, they represent a critical failure of faithfulness. Our goal is to ensure that each step is proven only taking into account the correct set of previous lemmas and theorem conditions specified by the original proof's logic. Enforcing this correct dependency structure offers significant practical advantages as well. It improves efficiency by constraining the autoformalizer's search space, which can lead to fewer spent tokens and faster verification.

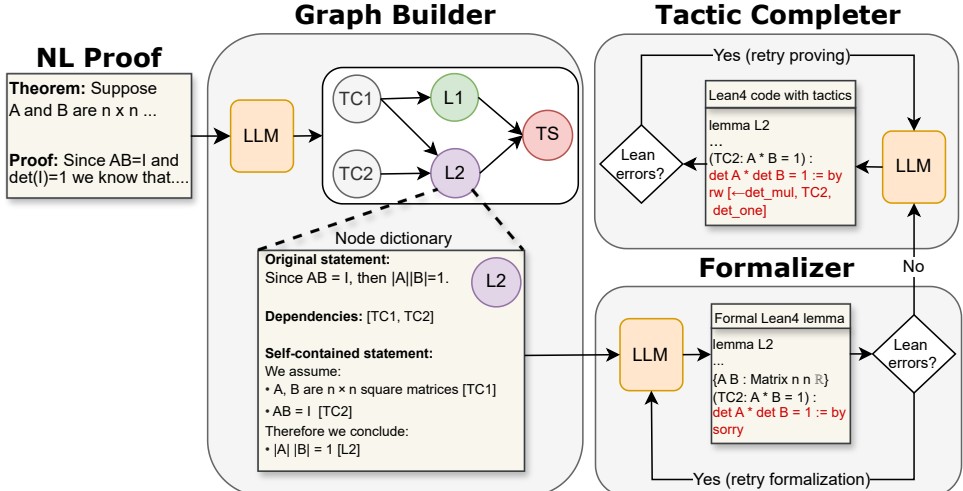

Figure 3: Our proof autoformalization pipeline with three parts: (1) Graph builder; (2) Lemma Formalizer; and (3) Tactic Completer. Lean errors are verified by the Lean 4 compiler.

### 3.1 New Workflow for Proof Autoformalization

As shown in Figure 3, our method enforces a correct dependency graph through a three-stage pipeline, leveraging LLMs at each step to bridge the gap between natural language (NL) and formal proof code. The first stage, Graph Builder, constructs a dependency directed acyclic graph (DAG) from the original proof. Next, the Formalizer uses an LLM to translate each proof step into formal Lean code. Finally, the Tactic Completer fills in the necessary tactics to complete the Lean proof. The specific LLM models used at each stage are detailed in Section 6.

**1. Graph Builder:** This step parses the natural language (NL) theorem and its proof to construct a dependency graph with LLM. Formally, this graph is a DAG, $G = (V, E)$, where $V$ is the set of all nodes and $E \subseteq V \times V$ is the set of directed edges. The nodes are partitioned into disjoint sets: $V = V_{TC} \cup V_D \cup V_L \cup V_{TS}$, representing Theorem Conditions, Definitions, Lemmas, and Theorem Solutions, respectively. Each edge $(u, v) \in E$ signifies that node $u$ is a prerequisite for proving the statement of node $v$. From the theorem statement, we extract nodes for theorem conditions and theorem solutions. From the proof statement, we extract nodes for lemmas and extra definitions. Each node is assigned its original NL statement, its dependencies, and a self-contained NL statement that provides a complete description of the current proof step (see Figure 3). To ensure the graph's validity, the system checks for forward references and cycles and verifies that every node, except the theorem solutions, has an outgoing edge. If the check fails, we task the LLM to improve the graph.

**2. Formalizer:** For each node in the graph, the LLM formalizes its self-contained NL statement into Lean 4 code. This process is iterative: generated errors are fed back to the LLM for correction. At this stage, each lemma is finalized with the "by sorry" placeholder, as tactics are not yet applied.

**3. Tactic Completer:** The final step completes the proofs for the lemmas by replacing the "by sorry" placeholders in the formalized Lean 4 code with the appropriate Lean 4 tactics.

To streamline the entire workflow illustrated in Figure 3, we developed a user-friendly Python package for PROOFFLOW. This package automates the complete process, allowing users to provide an informal theorem and proof with just a few lines of code. The package then autonomously executes the full workflow. It also integrates PROOFSCORE evaluation and our PROOFFLOWBENCH benchmark for comprehensive assessment. The package is available at `https://github.com/Huawei-AI4Math/ProofFlow`, with all the LLM prompts used in this project included in the repository.

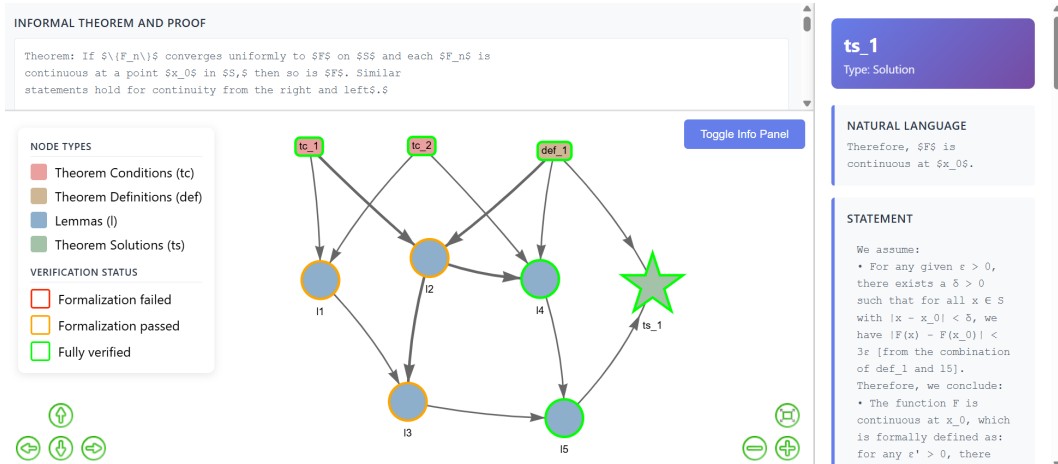

Figure 4: An example of the interactive visualization generated by PROOFFLOW. Node contours signify the outcome of each step: Red for a formalization error, orange for formalized statement without Lean 4 tactics, and green for formalized statement with Lean 4 tactics.

The final output of our package is an interactive diagram that visualizes the proof graph, as seen in Figure 4. Users can click on each node to get detailed information about its results. The contours of each node provide an at-a-glance summary of the outcomes, indicating whether its formalization and solving steps were successful. This allows users to immediately assess the progress of the autoformalization and pinpoint the locations in the proof graph where manual effort is needed.

## 4    SCORING PROOF AUTOFORMALIZATION AND ERROR DETECTION

As discussed in Section 1, a faithful autoformalization must satisfy three key properties: (1) Structural Fidelity, which ensures the proof's dependency graph is preserved; (2) Syntactic Correctness, which ensures the output is verifiable code without compilation errors; and (3) Semantic Faithfulness, which ensures each formalized statement accurately preserves the precise mathematical meaning of its original natural language statement.

### 4.1    PROOFSCORE

To evaluate the effectiveness of our PROOFFLOW pipeline, we introduce PROOFSCORE, a single unified score that synthesizes these three criteria.

**Structural Fidelity** is evaluated by checking whether the dependencies for a given node are valid. For a node $v_i$, we check if its estimated dependencies, $D_{\text{est}}(v_i) = \{u \in V_{\text{est}} \mid (u, v_i) \in E_{\text{est}}\}$, match the dependencies in a ground truth graph. We permit several valid graphs ($\mathcal{G}_{\text{true}}$) because the level of granularity can vary. For example, the calculation $1 + 13 + 5 = 14 + 5 = 19$ could be broken down into either one or two steps, depending on the user's granularity preference.

**Syntactic Correctness** is denoted by $c_i \in \{0, 1\}$, where $c_i = 1$ if the formalization of node $v_i$ is free of Lean 4 compilation errors at the end of the Tactic Completer step and $c_i = 0$ otherwise.

**Semantic Faithfulness** is assessed by adapting the "LeanScore" metric from Yu et al. (2025a), which was originally designed to evaluate the semantic faithfulness of a theorem statement formalization. A detailed description is given in Appendix A.2. This metric provides a faithfulness score, $f_i \in [0, 1]$, for each node $v_i$ of the dependency graph, and this score measures the semantic equivalence between the input NL statement and its corresponding formalized Lean 4 lemma. The higher the score, the more faithful the Lean 4 lemma is to the input NL statement.

The final unified PROOFSCORE, for a proof with $n$ steps is computed as:

$$\text{PROOFSCORE} = \frac{1}{n} \sum_i^n f_i\, c_i\, I[D_{\text{est}}(v_i) \in \mathcal{D}_{\text{true}}(v_i)],$$

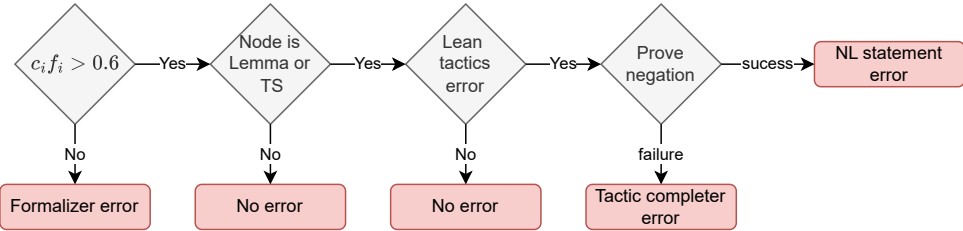

Figure 5: Flowchart illustrating the error detection mechanism. This is applied lemma by lemma and $f_i$ is the semantic faithfulness of node $v_i$ and $c_i = 0$ if there are syntactic errors.

where ProofScore $\in [0, 1]$, and $\mathcal{D}_{\text{true}}(v_i)$ is the set of valid dependencies for node step $v_i$. When the GraphBuilder is configured to enforce the DAG structure, structural fidelity is guaranteed for all nodes. Otherwise, we use the Claude-Sonnet-4 LLM with a specialized prompt to verify structural fidelity. This same LLM model is also used to check for semantic faithfulness, utilizing prompts available within the PROOFFLOW package. Additional details about PROOFSCORE, including examples of natural language proofs paired with their Lean code and respective scores, are provided in Appendix A.2.

## 4.2 ERROR ANALYSIS

PROOFFLOW also has an error detection system operating at each proof step. This process (Figure 5) determines the error source, which can be the formalizer, tactic completer, or original natural language proof. First, a proof node's semantic faithfulness score is compared against a 0.6 threshold, following Yu et al. (2025a). If it passes, the system checks if it's a provable statement (lemma or theorem solution). Otherwise, the process for that node ends. For provable statements, the tactic completer attempts to apply Lean 4 tactics. Success means the proof step is complete with no errors. If the tactics fail, the LLM tries to prove the negation of the statement. If the negation can be proven, then the original natural language statement is deemed imprecise. Otherwise, the error is considered to be with the tactic completer LLM, which is incapable of either proving or disproving the step.

## 5 PROOFFLOWBENCH

Existing mathematical benchmarks are often limited to pure calculation and are designed to test problem solving accuracy. Previous natural language proof datasets are also not self-contained, often referencing external sources for non-standard theorems (Welleck et al., 2021). Furthermore, they focus on specific topics and sometimes also including calculation problems instead of proofs (Sheng et al., 2025). To address these limitations, we introduce a new benchmark, PROOFFLOWBENCH, to specifically evaluate automated proof formalization pipelines. The benchmark, provided as part of the PROOFFLOW package, consists of 184 undergraduate-level mathematics theorems and proofs in natural language from six key areas: number theory and algebra (27), real analysis (42), inequality (36), probability and set theory (31), complex analysis (25), and sequences and series (23). To construct this dataset, we adapted 63 problems from the NaturalProofs benchmark (Welleck et al., 2021) and 36 problems from the IneqMath benchmark (Sheng et al., 2025).

The benchmark also contains the natural language proofs divided into proof steps, and the respective dependency graphs (DAGs), which can be used to evaluate structural fidelity. On average, each problem's graph consists of 8.4 total nodes (more statistics provided in Appendix A.3). The proof graphs have been manually validated. However, we emphasize that for the same proof different proof graphs are possible depending on the desired level of detail of each proof node.

## 6 COMPARISON STUDY

To evaluate our PROOFFLOW pipeline, we conduct a comparison study with three main objectives. First, we assess the effectiveness of formalizing proofs using our high-level, lemma-based approach (PROOFFLOW) versus low-level tactics. This compares the "Lemma Formalization" (ours) with the "Tactic Formalization" (existing), as depicted in Figure 1. Secondly, to examine the role of explicit

dependency management in ensuring structural fidelity to the original proof, we compare the DAG-enforcing version of PROOFFLOW with a variant where this is relaxed (noDAG). Finally, we conduct the error analysis of Section 4.2, to detect the source of errors in the PROOFFLOW pipeline variants.

## 6.1 EXPERIMENTAL SETTINGS

**PROOFFLOW**: We consider the PROOFFLOW variant that explicitly enforces the correct dependency graph as our main method, referred to as PROOFFLOW DAG. To evaluate the role of this dependency enforcement, as an ablation study, we also consider a PROOFFLOW noDAG version, where such mechanism is relaxed. That is, in PROOFFLOW noDAG, all previous lemmas and premises are provided when formalizing each step, similarly to how dependency is handled by most prior work.

**Existing methods**: We compare our pipeline against existing tactic-based formalization methods: (1) "FULL PROOF" autoformalization (Lu et al., 2024), which calls the LLM once to formalize the entire theorem and proof using Lean 4 tactics, and (2) "STEP PROOF" autoformalization (Hu et al., 2025), which formalizes one proof step at a time into Lean 4 tactics. We note that STEP PROOF utilizes the same Graph Builder LLM and prompt as PROOFFLOW to decompose the initial NL proof into steps.

**Thinking modes**: All methods were evaluated in both thinking and non-thinking modes to provide a more comprehensive assessment. For existing methods, Goedel-Prover-V2-32B and Gemini-Flash-2.5 (Gemini Team, 2025) are used for the thinking and non-thinking modes. For our PROOFFLOW pipeline, the Graph Builder always uses Gemini-2.5-Pro; the thinking mode utilizes Goedel-Formalizer-2-32B and Goedel-Prover-V2-32B (Lin et al., 2025) as the Formalizer and the Tactic Completer, while non-thinking mode utilizes Gemini-2.5-Flash and DeepSeekProver V2 671B (Ren et al., 2025) for the two components. All LLM models were given appropriate system prompts.

**Evaluation metrics:** Our main evaluation metric in PROOFSCORE, supplemented by measures of syntactic correctness, both at step-level and proof-level. Step-level syntactic correctness of the generated Lean 4 code is measured at two key stages: after the Formalizer step (Formalizer accuracy) and after the Tactic Completer step (Tactic accuracy). We also report proof-level "correct syntax", whether all of the formalizer and solver steps are syntactically valid. All syntactic correctness checks are verified by LeanServer (Santos et al., 2025) with Lean version v4.15.0. We also adapted PROOFSCORE (Sect. 4), so it could evaluate the FULL PROOF and STEP PROOF pipelines (see Appendix A.2.2 for details). Finally, we report the elapsed time and the number of generated tokens. Since our experiments use both local and API-based model deployments, inference speeds vary. Therefore, the number of generated tokens is a more meaningful indicator of the effort required.

**Data and reproducibility**: We utilized the problems in the PROOFFLOWBENCH benchmark for this comparison study. Reproducible code is provided as part of the PROOFFLOW package.

## 6.2 EMPIRICAL RESULTS

The results evaluated at a Pass@5 rate are presented in Table 1. This Pass@5 setting allows for up to five self-correction retries for each stage in the pipelines. If a generated step produces syntactically incorrect code, the model is re-prompted with the error and its previous output, giving it an opportunity to fix the mistake. The results for Pass@1 and 3 are show in Appendix A.4.

**Existing Methods Comparison:** The results clearly demonstrate that our pipeline outperforms existing methods (FULL PROOF and STEP PROOF), achieving the highest PROOFSCORE (0.545). The FULL PROOF method calls the LLM once for the entire proof formalization and therefore we only report proof-level results. This approach achieves a comparatively high syntax passing rates (0.571 for the thinking mode), however, the PROOFSCORE was comparatively low, indicating that the generated formal proofs, while syntactically correct, often fail to faithfully capture the structure and intent of the original natural language proof. The STEP PROOF method exhibits the lowest proof-level syntax passing rates (0.005 for the non-thinking mode and 0.119 for the thinking mode) because the LLM struggles to maintain correct and consistent indentation in Lean 4 across steps. Also, once an error appears in one step, subsequent steps cannot achieve syntactic correctness. The poor syntax passing rates of the STEP PROOF approach results in correspondingly low PROOFSCORE values, given that this metric incorporates syntactic correctness as a component.

Table 1: Performance metrics under the Pass@5 setting on our 184-problem benchmark. Step-level averages are computed over all individual steps, while proof-level averages are computed per proof. Entries marked with "/" indicate not applicable.

| | | Step-Level | | Proof-Level | | | |
| | Think | Form. | Tactic | Proof | Correct | Time | Output |
| Pipeline | mode | accuracy | accuracy | Score | syntax | (mins) | tokens (k) |
|---|---|---|---|---|---|---|---|
| **PROOFFLOW** | No | 0.751 | 0.358 | 0.355 | 0.027 | 8.8 | 22.4 |
| **DAG** | Yes | **0.939** | **0.742** | **0.545** | 0.375 | 31.8 | 94.2 |
| **PROOFFLOW** | No | 0.807 | 0.391 | 0.347 | 0.049 | 12.3 | 25.8 |
| **noDAG** | Yes | 0.936 | 0.681 | 0.417 | 0.353 | 32.0 | 98.5 |
| **FULL PROOF** | No | / | / | 0.021 | 0.027 | 0.8 | 10.5 |
| | Yes | / | / | 0.279 | **0.571** | 3.8 | 15.1 |
| **STEP PROOF** | No | / | 0.068 | 0.046 | 0.005 | 0.2 | 1.2 |
| | Yes | / | 0.129 | 0.029 | 0.119 | 10.6 | 32.9 |

**Ablation Study**: The PROOFSCORE evaluation metric establishes the DAG version of PROOFFLOW as the top-performing method for proof autoformalization. This is most evident in the "thinking" mode, where the DAG variant achieves a PROOFSCORE of 0.545, compared to the noDAG's 0.417, due to improved structural fidelity and syntactic correctness. While the DAG also outperforms the noDAG variant for all syntax metrics in the thinking mode, we observed a higher syntax passing rate for the noDAG in the non thinking setting. This is because the noDAG variant provides all previous steps as known conditions to the LLM, and this extra information can lead to higher syntactic passing rates for weaker models, but often results in a logically inconsistent formal proof structure, as shown by the lower PROOFSCORE averages. This finding underscores the critical importance of maintaining structural fidelity.

**Alternative benchmark:** We additionally validated our findings on the established miniF2F benchmark (Zheng et al., 2021) by evaluating 50 randomly selected test problems. The results (Appendix A.7) confirm that the relative performance ordering across all pipeline configurations is preserved, with all accuracy indicators universally higher on miniF2F, indicating it represents an easier evaluation setting.

**Computational efficiency**: Appendix A.8 provides a detailed breakdown of computational costs across pipeline components. Our analysis reveals that the Tactic Completer stage is the primary computational bottleneck, consuming 81–89% of total execution time across all configurations. A straightforward solution to address this bottleneck, enabled by our DAG-based pipeline design, is parallelization. Our architecture naturally supports parallel execution at multiple levels: proof nodes in different branches can be formalized concurrently, and once formalized, tactic completion can proceed independently across nodes. In principle, our DAG design enables extensive parallelization that could substantially reduce wall-clock time. However, we did not implement parallelization in our experiments due to limited GPU resources, which restricted us to sequential processing.

## 6.3 ERROR ANALYSIS

The error analysis, shown in Table 2, identifies the Formalizer as the primary source of failure, accounting for 32% to 47% of all autoformalization outcomes, depending on the pipeline configuration. These failures are predominantly semantic, stemming from discrepancies between the natural language and the formalized Lean 4 code, despite the high syntactic correctness (Table 1). Our most robust configuration, which uses the PROOFFLOW DAG architecture in thinking mode, has 53.3% error-free proof steps and clearly outperforms the 42.8% rate of the baseline (noDAG) version. This confirms that while the DAG architecture improves performance, improving semantic preservation during the formalization stage remains a challenge for future work.

Table 2: Breakdown of step-level outcomes for different pipeline configurations. "None" indicates the percentage of total steps completed successfully, while other columns show the percentage of steps that failed due to a specific error source.

| Pipeline | Think | Total Steps | Error Source (%) | | | |
|---|---|---|---|---|---|---|
| | | | None | Formalizer | Tactic | NL Statement |
| PROOFFLOW DAG | No | 1735 | 33.7 | 46.5 | 19.6 | 0.2 |
| PROOFFLOW DAG | Yes | 1737 | **53.3** | 38.9 | 5.6 | 2.2 |
| PROOFFLOW noDAG | No | 1751 | 32.8 | 45.6 | 21.4 | 0.2 |
| PROOFFLOW noDAG | Yes | 1755 | 42.8 | 47.0 | 7.6 | 2.6 |

The PROOFFLOW's error detection process goes beyond simple checks, being able, in some instances, to identify a spectrum of flaws in the initial natural language proof. Examples are provided in Appendix A.6. These detected issues range from blatant algebraic errors to subtle ambiguities and unstated assumptions. By flagging these mistakes, pipelines like PROOFFLOW can provide feedback that not only helps correct errors but also strengthens the overall rigor and clarity of the proof.

## 7 DISCUSSION

Our results demonstrate the importance of preserving a proof's structural fidelity. The PROOFFLOW lemma-based approach significantly outperforms both monolithic ("FULL PROOF") and sequential ("STEP PROOF") methods, which falter by tackling excessive complexity at once or by imposing a linear structure unfaithful to the proof logic. Our method succeeds by using a lemma-based structure that explicitly models the proof's dependency graph. This approach deconstructs the problem into manageable, logically-constrained steps. It guides the LLM along the author's intended path, preventing logical "shortcuts" that undermine other approaches. However, performance hinges on the Formalizer step, where poor semantic preservation remains the primary bottleneck.

A limitation of our work was the exclusive focus on undergraduate-level proofs, omitting research-level problems. However, this scope reflects the current capabilities of the field rather than intrinsic limitations of the pipeline architecture we propose. Testing our system on more challenging research-level problems did reveal the expected performance drop, consistent with observations across the literature (e.g., (Jiang et al., 2025)). Critically, our pipeline's core contributions, the DAG-based lemma decomposition and structural preservation, are not the primary limiting factors. The main bottlenecks lie in the underlying formalizer and tactic-completer LLMs, which currently struggle with research-level concepts, and in MathLib's incomplete coverage of specialized mathematical domains. As these foundational components mature and the formal mathematics ecosystem grows, we anticipate that our pipeline's performance will naturally improve, providing a scalable framework that is designed to grow more capable alongside these broader advances.

Our modular three-stage pipeline also enables the curation of high-quality training data for both end-to-end proof autoformalization models and individual Formalizer and Tactic Completer LLMs, a key direction for future work. Our error detection mechanisms can filter low-quality autoformalizations at different pipeline stages, creating curated datasets that support multiple training paradigms: supervised fine-tuning on the complete chain-of-thought reasoning process and reinforcement learning using detected errors as negative examples.

Ultimately, our work demonstrates that proof autoformalization is an achievable goal: we achieved 37.5% proof-level syntactic accuracy on undergraduate problems, a significant leap from existing approaches that only achieved 6.10% accuracy on simpler middle-school mathematics (Hu et al., 2025). More importantly, we establish that for these tools to be truly useful to mathematicians, they must faithfully represent the structure and logic of the human-created arguments they seek to formalize. Addressing the current bottlenecks could lead to practical tools that function as a proof "auto-correct" for mathematicians, pinpointing errors and ambiguities directly in natural language and in real time.

ETHICS STATEMENT

This research adheres to the ICLR Code of Ethics and does not involve human subjects, privacy concerns, or ethical issues. The study uses only publicly available mathematical data, with no personal or sensitive data involved.

The proposed PROOFFLOW pipeline, PROOFSCORE metric, code, and datasets are designed to assist mathematicians in theorem formalization, aiming to advance mathematical research and AI for mathematics. This work poses no foreseeable harmful applications and contributes positively to the scientific community by enhancing automated theorem proving capabilities.

REPRODUCIBILITY STATEMENT

To ensure the reproducibility of this work, we have made all necessary materials openly available. The complete implementation of the methodology (PROOFFLOW pipeline, PROOFSCORE metric), all experimental data, LLM prompts and comparison study code are publicly accessible through our package repository: `https://github.com/Huawei-AI4Math/ProofFlow`. The provided GitHub repository includes detailed instructions and documentation for replicating all experimental results presented in this paper.

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

# A    APPENDIX

## A.1    LLM USAGE STATEMENT

In this work, Large Language Models (LLMs) were employed in specific components of our research pipeline and experimental framework. The primary applications of LLMs include:

1. **PROOFFLOW Pipeline Implementation**: LLMs were utilized as core components within our proposed ProofFlow pipeline as Graph Builder, Formalizer and Tactic Completer.

2. **Existing Methods Comparison**: During the experimental evaluation, LLMs were employed to perform inference with existing methods for comparative analysis against our proposed approach.

3. **PROOFSCORE Computation**: LLMs were integrated into the calculation process of our proposed PROOFSCORE metric for quantitative assessment of formalization quality.

4. **Benchmark Dataset Construction**: The proof graphs included in our benchmark dataset were initially generated by LLMs, followed by human verification.

LLMs served as computational tools in these specific applications and for grammatical correction of an early draft. All research ideation, methodological design, analysis, and interpretation of results were conducted exclusively by the human authors, who take full responsibility for the content of this paper, including any LLM-assisted components.

## A.2    ADDITIONAL DETAILS ON PROOFSCORE

### A.2.1    EVALUATION OF SEMANTIC EQUIVALENCE OF PROOFSCORE

In this section, we detail how to obtain the semantic equivalence score $f_i$ of each node, as proposed in Section 4. Note that by default, the score $f_i$ is 0 if the syntactic check at the formalizer step is not passed.

In particular, we assess the semantic equivalence between the self-contained natural language statement $nls_i$ and Lean code $lc_i$ of each node, which are the input and output of the Formalizer step of PROOFFLOW (Sect. 3.1). The process is as follows:

1. First, we prompt LLM to break down both $nls_i$ and $lc_i$ into components.

2. We employ an LLM-as-a-judge to evaluate the semantic equivalence between each formalized component, $lc_i$, and its natural language counterpart, $nls_i$. The LLM is provided with few-shot examples illustrating specific error types and assigns one of the following evaluations to each component:

   - *Perfectly match*: The component $lc_i$ fully captures the reasoning of $nls_i$. It may include additional constraints or conditions that were implicit in the natural language but are necessary for formal rigor.
   - *Minor inconsistency*: The component $lc_i$ correctly represents the core logic of $nls_i$ but may feature slight structural reordering or other small deviations.
   - *Major inconsistency*: The component $lc_i$ either omits a key part of the logic from $nls_i$ or introduces entirely different reasoning.

3. A Fuzzy measurement score is then computed for each component based on these evaluations, following the method in Yu et al. (2025a). This score is effectively a weighted average of the counts for "Perfectly match" and "Minor inconsistency," with zero tolerance for any component rated as a "Major inconsistency."

4. The final score, $f_i$, is calculated by aggregating the Fuzzy measurements using a Sugeno Integral (Sugeno, 1974).

In brief, the score $f_i$ of a proof step is equal to $1$ if the Lean code fully captures the self-contained natural language statement with all necessary conditions and $f_i$ is equal to $0$ if at least one component of the Lean code contains major flaws.

### A.2.2 PROOFSCORE FOR FULL PROOF AND STEP PROOF

Our evaluation score, PROOFSCORE, is the average of faithfulness evaluations of individual steps (nodes in the dependency graph) of our PROOFFLOW method. To create a fair comparison with FULL PROOF and STEP PROOF, we adapted this evaluation metric as follows. In all cases, any syntactically incorrect Lean code was automatically assigned a score of 0.

- For FULL PROOF: Since this method generates a single, complete proof for each problem, we applied the scoring process (detailed in Appendix A.2.1) to the entire proof. The final score in Tables 1 and 6 is the average score across all problems. Note that if the proof fails Lean syntactic check, the score is 0 by default.
- For STEP PROOF: This method generates code for each individual step. Therefore, we applied the scoring process to each step independently. The final score in the tables is the average across all individual steps from every problem in the benchmark. If any step fails Lean syntactic check, the score is 0 by default.

The results of this comparative analysis are presented in Tables 1 and 6.

### A.2.3 RELIABILITY AND CONSISTENCY OF PROOFSCORE

As Claude is not involved in any module of the pipeline (Graph builder, Formalizer and Tactic Completer), we employ Claude-Sonnet-4 to evaluate the semantic faithfulness in PROOFSCORE.

To further validate the reliability of PROOFSCORE, we compare it with 2 variants (using Gemini-2.5-Flash and DeepSeek-V3 in lieu of Claude) and merely Claude-as-a-Judge. While the variants of PROOFSCORE evaluate faithfulness on a component basis and aggregate using Sugeno Integral (see Sect. 4), for Claude-as-a-Judge, we prompt it to evaluate directly whether the Lean code is semantically equivalent to the natural language proof step.

We employ these methods to evaluate a sample of 200 natural language proof steps from our benchmark and check their formalization faithfulness evaluations against the results obtained by a human expert. For human evaluations, the ground-truth label is "PASS" if the Lean code is semantically equivalent to natural language and "FAIL" otherwise. Consistently with Section 6, for variants of PROOFSCORE, a proof score that is larger or equal to 0.6 means the scorer method predicts "PASS", and the scorer predicts "FAIL" otherwise. On the other hand, Claude-as-a-Judge predicts "PASS"/"FAIL" directly. We compute the F1 scores of these predictions versus the ground-truth labels.

The results are summarized in Table 3. While the variants of PROOFSCORE obtain close performances, they all achieve higher scores in judging semantic equivalence than Claude-as-a-Judge. These results suggest that PROOFSCORE is reliable in judging semantic faithfulness and this reliability is consistent across different Large Language Models.

Table 3: Comparison of F1 scores of variants PROOFSCORE by Claude, Gemini and DeepSeek and Claude-as-a-Judge against human expert evaluations. All variants of PROOFSCORE achieve a higher level of correctness in judging semantic equivalence than Claude, demonstrating the reliability and consistency of PROOFSCORE.

| Method | Claude-as-a-Judge | PROOFSCORE Claude (paper) | PROOFSCORE Gemini | PROOFSCORE DeepSeek |
|---|---|---|---|---|
| F1-score | 0.83 | 0.91 | 0.88 | 0.9 |

### A.2.4 EXAMPLES OF PROOFSCORE RESULTS

To illustrate how PROOFSCORE behaves for each proof node, we provide several examples in Tables 4 and 5. Each example shows a natural language proof step paired with its formalized Lean code and the corresponding PROOFSCORE. Note that we display each proof step's individual contribution to the overall PROOFSCORE from Section 4—that is, the product of terms within the summation (each term in the sum).

In short, any Lean code is assigned a score of 0 if either the code is syntactically incorrect or the tactics are structurally unfaithful to the proof step or there is at least one component that is significantly inconsistent with the natural language. By contrast, score 1 means that all components of the Lean code match perfectly with their counterparts in the natural language proof step.

| Natural Language | Lean code | Score and Explanation |
|---|---|---|
| From $n = 2k + 1$ [l1] and $n^2 = 4k^2 + 4k + 1$ [l2], we conclude: $n^2 = 4k(k+1)+1$ [l3]. (Example highlighted in Figure 2(b)). | ```lemma l3 (n : Z) (tc_1 : n % 2 = 1) (l1 : $\exists$ k : Z, n = 2*k + 1) (l2 : n^2 = 4 * (Classical. choose l1)^2 + 4 * ( Classical.choose l1) + 1) : n^2 = 4 * (Classical.choose l1 ) * ((Classical.choose l1) + 1) + 1 := by have h : n = 2 * (Classical. choose l1) + 1 := by have h1 : n = 2 * ( Classical.choose l1) + 1 := by exact Classical.choose_spec l1 exact h1 have h3 : n ^ 2 = 4 * ( Classical.choose l1) ^ 2 + 4 * (Classical.choose l1) + 1 := by have h4 : n ^ 2 = 4 * ( Classical.choose l1) * (( Classical.choose l1) + 1) + 1 := by have h5 : 4 * (Classical. choose l1) ^ 2 + 4 * ( Classical.choose l1) + 1 = 4 * (Classical.choose l1) * ((Classical.choose l1) + 1) + 1 := by ring_nf linarith [h3, h5] exact h4 ``` | Score 0: structurally unfaithful. The code to prove L3 only utilizes header by L1 while ignoring L2, structurally unfaithful to the input NL proof. |
| We have: $(x-2)^2+(y-1)^2 = 5$ and $(x-4)^2+(y-1)^2 = 5$. So, $x = 3$. | ```theorem l3 (x y : R) (l1 : (x - 2)^2 + (y - 1)^2 = 5) (l2 : (x - 4)^2 + (y - 1)^2 = 5) : x = 3 = by sorry ``` | Score 0: wrong syntax. It should be ":= by sorry" but "= by sorry" was used. As the code fails Lean check (syntax error), the PROOF-SCORE is 0. |

Table 4: Examples of varying PROOFSCORE with natural language proof step and Lean code.

## A.3 BENCHMARK DATASET STATISTICS

The distribution of the number of nodes per proof in the PROOFFLOWBENCH benchmark is shown in Figure 6. On average, the proofs have 2 theorem conditions, 0.6 definitions, 4.4 lemmas, and 1.2 theorem solutions.

| Natural Language | Lean code | Explanation |
|---|---|---|
| Because $(a+b-c), (b+c-a), (c+a-b) > 0$ and we have: $a^2 \geq (a+c-b)(a+b-c), b^2 \geq (b+c-a)(b+a-c)$, and $c^2 \geq (c+b-a)(c+a-b)$. Thus: $a^2b^2c^2 \geq (a+c-b)(a+b-c)(b+c-a)(b+a-c)(c+b-a)(c+a-b)$. | ```theorem l6 (a b c : R) (tc_1 : a > 0 ^ b > 0 ^ c > 0 ^ a + b > c ^ b + c > a ^ c + a > b) (l4 : (a + b - c > 0) ^ (b + c - a > 0) ^ (c + a - b > 0)) (l5 : a^2 >= (a + c - b) * (a + b - c) ^ b^2 >= (b + c - a) * (b + a - c) ^ c^2 >= (c + b - a) & (c + a - b)) : a^2 * b^2 * c^2 > (a + c - b) * (a + b - c) * (b + c - a) * (b + a - c) * (c + b - a) * (c + a - b) := by sorry``` | Score 0: major inconsistency. The Lean code is syntactically correct but contains a major inconsistency at the proposition. While NL has $\geq$, the proposition in Lean has $>$. This results in Proof Score 0. |
| As the complex conjugate of $z$ is $\bar{z} = x - iy$, we conclude: $(2-i)\bar{z} = (2x-y) - i(x+2y)$. | ```theorem l3 (x y : E) (z : C) ( h : z = x + y * Complex.I) : (2 - Complex.I) * star z = (2 * x - y) - Complex.I * (2 * y + x) := by sorry``` | Score 0.67: minor inconsistency. The Lean code is syntactically correct and equivalent to the proof. The right-hand side of the proposition contains a slight variable ordering: $(x+2y)$ vs. $(2*y+x)$ in Lean. However, this change is minor and still preserves the semantic meaning of the proof step. |
| Because $(a+b-c), (b+c-a), (c+a-b) > 0$ and we have: $a^2 \geq (a+c-b)(a+b-c), b^2 \geq (b+c-a)(b+a-c)$, and $c^2 \geq (c+b-a)(c+a-b)$. Thus: $a^2b^2c^2 \geq (a+c-b)(a+b-c)(b+c-a)(b+a-c)(c+b-a)(c+a-b)$. | ```theorem l6 (a b c : R) (tc_1 : a > 0 ^ b > 0 ^ c > 0 ^ a + b > c ^ b + c > a ^ c + a > b) (l4 : (a + b - c > 0) ^ (b + c - a > 0) ^ (c + a - b > 0)) (l5 : a^2 >= (a + c - b) * (a + b - c) ^ b^2 >= (b + c - a) * (b + a - c) ^ c^2 >= (c + b - a) & (c + a - b)) : a^2 * b^2 * c^2 >= (a + c - b) * (a + b - c) * (b + c - a) * (b + a - c) * (c + b - a) * (c + a - b) := by sorry``` | Score 1: perfectly match. The Lean code is syntactically correct and semantically equivalent to the proof step. |

Table 5: Examples of varying ProofScores with natural language proof step and Lean code.

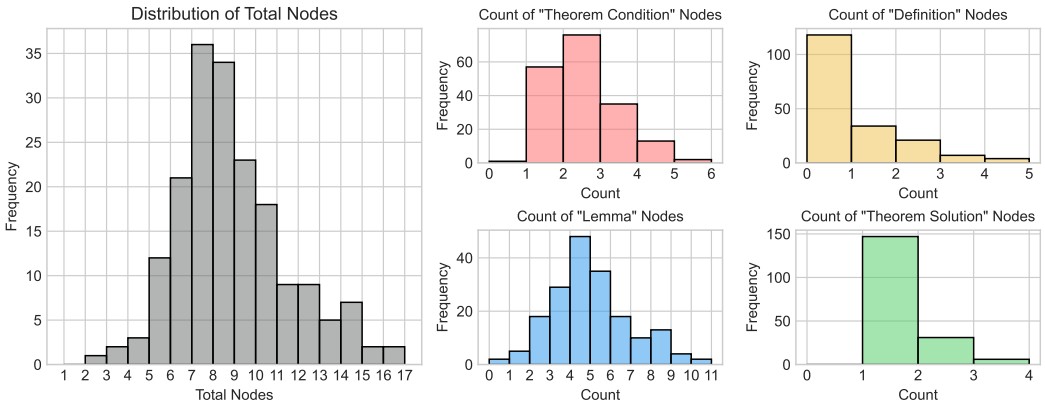

Figure 6: The distribution of the total number of nodes per problem (left) and the frequency of each node type (right).

### A.4 COMPARISON RESULTS

This section contains extra comparison results. We show in Table 6 the extended version of Table 1, which includes Pass@1, 3, and 5 rates.

### A.5 COMPARISON EXAMPLES

In this section, we present several examples of proof autoformalization to illustrate how different pipelines perform when formalizing natural language proofs. The examples below, selected from our comparative study in Section 6, show how PROOFFLOW DAG maintains a high degree of fidelity to the input proof. In contrast, other approaches, such as PROOFFLOW noDAG, FULL PROOF, and STEP PROOF, often fail to either adhere to the flow of the natural proof or even generate valid Lean 4 code.

To evaluate structural fidelity, we analyzed the dependencies for each syntactically correct proof step. By inspecting the Lean tactics, we identified exactly which previously proven steps and theorem conditions the solver utilized. A step was deemed structurally faithful if this set of dependencies precisely matched the logic of the original natural language proof. If the dependencies differed in any way, the step was marked as unfaithful.

For the purposes of illustration, the figures presented in this section are based on the actual dependency graph of the input natural language proof, and annotations will be provided in the case the the logical flow of the generated proof graph deviates from this input dependency graph.

#### A.5.1 EXAMPLE: PROOFFLOW DAG IS SUPERIOR IN STRUCTURE FIDELITY

This example corresponds to entry "dummy_6" in the PROOFFLOWBENCH benchmark.

> **Theorem**
>
> If $n$ is an odd integer, then $n^2 \equiv 1 \pmod 8$.

> **Proof**
>
> Since $n$ is odd, we can write $n = 2k + 1$ for some integer $k$. Then $n^2 = (2k + 1)^2 = 4k^2 + 4k + 1$. We can factor this as $n^2 = 4k(k + 1) + 1$. Now, either $k$ is even or $k$ is odd. If $k$ is even, then $k + 1$ is odd, and if $k$ is odd, then $k + 1$ is even. In either case, $k(k + 1)$ is even, so $k(k + 1) = 2m$ for some integer $m$. Therefore $n^2 = 4(2m) + 1 = 8m + 1$, which means $n^2 \equiv 1 \pmod 8$.

Table 6: Performance metrics for all pipelines, evaluated at Pass@1, 3, and 5 rates on our 184-problem benchmark. Entries marked with "/" indicate not applicable.

| Pipeline | Think mode | Pass | Step-Level | | Proof-Level | | | |
| | | | Form. accuracy | Tactic accuracy | Proof Score | Correct syntax | Time (mins) | Output tokens (k) |
|---|---|---|---|---|---|---|---|---|
| PROOFFLOW DAG | No | 1 | 0.644 | 0.252 | 0.320 | 0.016 | 3.3 | 8.1 |
| | | 3 | 0.722 | 0.323 | 0.347 | 0.027 | 6.3 | 15.7 |
| | | 5 | 0.751 | 0.358 | 0.355 | 0.027 | 8.8 | 22.4 |
| | Yes | 1 | 0.844 | 0.629 | 0.508 | 0.245 | 19.3 | 55.7 |
| | | 3 | 0.925 | 0.723 | 0.541 | 0.348 | 27.0 | 78.0 |
| | | 5 | **0.939** | **0.742** | **0.545** | 0.375 | 31.8 | 94.2 |
| PROOFFLOW noDAG | No | 1 | 0.697 | 0.225 | 0.312 | 0.022 | 4.8 | 9.8 |
| | | 3 | 0.791 | 0.328 | 0.344 | 0.038 | 9.0 | 18.4 |
| | | 5 | 0.807 | 0.391 | 0.347 | 0.049 | 12.3 | 25.8 |
| | Yes | 1 | 0.860 | 0.573 | 0.397 | 0.217 | 18.9 | 56.0 |
| | | 3 | 0.921 | 0.662 | 0.414 | 0.332 | 27.0 | 80.0 |
| | | 5 | 0.936 | 0.681 | 0.417 | 0.353 | 32.0 | 98.5 |
| FULL PROOF | No | 1 | / | / | 0 | 0 | 0.2 | 2.7 |
| | | 3 | / | / | 0.011 | 0.011 | 0.5 | 6.6 |
| | | 5 | / | / | 0.021 | 0.027 | 0.8 | 10.5 |
| | Yes | 1 | / | / | 0.156 | 0.309 | 0.9 | 3.5 |
| | | 3 | / | / | 0.216 | 0.467 | 2.5 | 9.5 |
| | | 5 | / | / | 0.279 | **0.571** | 3.8 | 15.1 |
| STEP PROOF | No | 1 | / | 0.054 | 0 | 0 | 0.03 | 0.1 |
| | | 3 | / | 0.061 | 0 | 0 | 0.05 | 0.3 |
| | | 5 | / | 0.068 | 0.046 | 0.005 | 0.2 | 1.2 |
| | Yes | 1 | / | 0.123 | 0.023 | 0.065 | 1.3 | 4.9 |
| | | 3 | / | 0.127 | 0.028 | 0.092 | 3.7 | 12.9 |
| | | 5 | / | 0.129 | 0.029 | 0.119 | 10.6 | 32.9 |

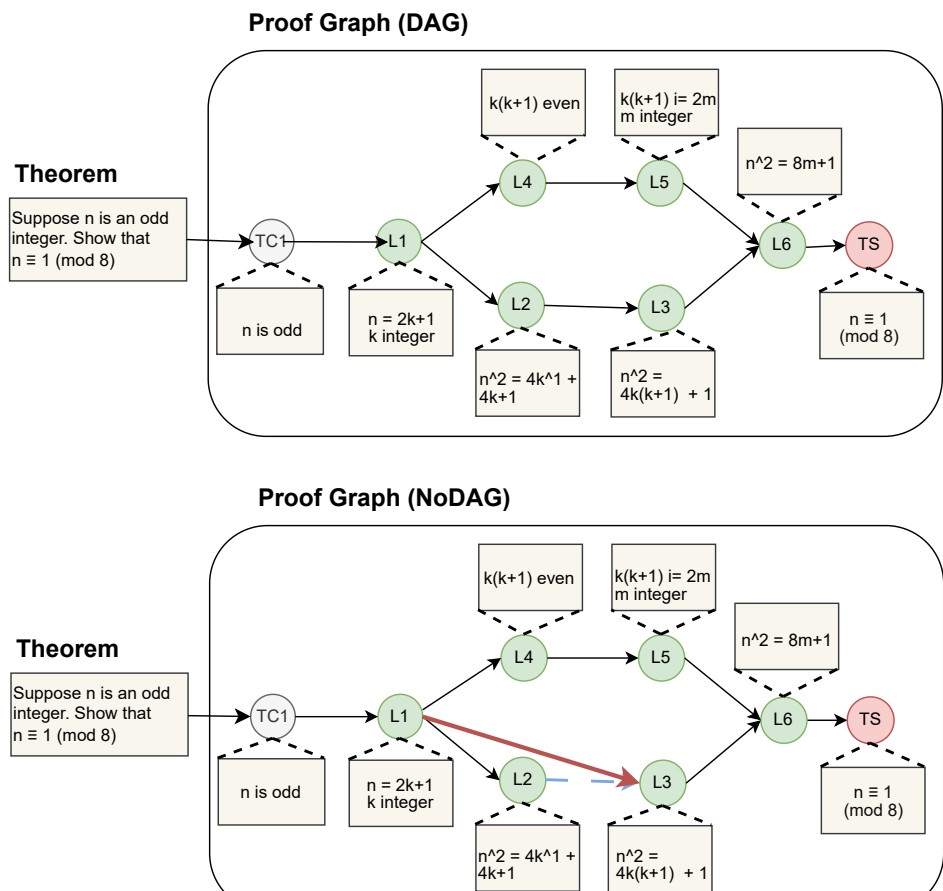

Figure 7: Comparison of proof structures generated by PROOFFLOW DAG (top) and PROOFFLOW noDAG (bottom) with respect to the original natural language proof. The red arrow indicates a dependency in PROOFFLOW noDAG not in the natural language proof. The blue dashed arrow indicates a dependence in the natural language proof but not included in PROOFFLOW noDAG. The PROOFFLOW DAG formal proof faithfully follows the dependency of the original proof. However, the one by PROOFFLOW noDAG proves step L2 then reuses L1 to prove L3, which is structurally unfaithful to the original proof.

**PROOFFLOW DAG:** The proof produced by PROOFFLOW DAG follows correctly the structure of the natural language proof and generates syntactically correct Lean code (see Figure 7).

**PROOFFLOW noDAG:** PROOFFLOW noDAG fails to adhere to the dependency structure of the natural language proof. In particular, as illustrated in Fig. 7, the transition from step L2 to L3 was inherent in the natural language proof but neglected in noDAG. The proof code for step L3 utilized step L1 to reprove the result of step L2. In other words, step L2 was made redundant.

**FULL PROOF:** FULL PROOF failed to generate syntactically correct Lean code (the first syntax error is "unknown identifier").

**STEP PROOF:** STEP PROOF failed to generate syntactically correct Lean code for steps L4, L5, L6, and TS (the first syntax error was "type mismatch").

### A.5.2 EXAMPLE: PROOFFLOWDAG IS SUPERIOR IN PROVER ACCURACY

This example corresponds to entry "dummy_7" in the PROOFFLOWBENCH benchmark.

> **Theorem**
>
> If $P(A) = 0.6$ and $P(B) = 0.7$, then $P(A \cap B) \geq 0.3$.

> **Proof**
>
> We know that $P(A \cup B) = P(A) + P(B) - P(A \cap B)$. Since $P(A \cup B) \leq 1$, we have $P(A) + P(B) - P(A \cap B) \leq 1$. Substituting the given values: $0.6 + 0.7 - P(A \cap B) \leq 1$, which gives $1.3 - P(A \cap B) \leq 1$. Therefore $P(A \cap B) \geq 0.3$.

Figure 8: The structure of the proof produced by PROOFFLOW DAG faithfully follows the dependency graph of the input natural language proof.

**PROOFFLOW DAG:** PROOFFLOW DAG generates step-by-step proof faithfully following the logical transition of the natural language proof (see Fig. 8).

**PROOFFLOW noDAG:** The Lean code proof achieves very low prover accuracy. This is likely attributed to the fact that each step is given all previous steps, in contrast to only the necessary steps. As a consequence, the prover fails to produce concise and correct Lean code.

**FULL PROOF:** FULL PROOF failed to generate syntactically correct Lean code to prove this theorem (the first syntax error is "unknown identifier").

**STEP PROOF:** STEP PROOF failed to generate syntactically correct Lean code to prove this theorem (the first syntax error is function type error).

### A.5.3    EXAMPLE: PROOFFLOW DAG IS SUPERIOR IN PROOF EFFICIENCY

This example corresponds to entry "dummy_9" in the PROOFFLOWBENCH benchmark.

> **Theorem**
>
> If $(a_n)$ is an arithmetic sequence with $a_1 = 5$ and $a_3 = 11$, then $a_5 = 17$.

> **Proof**
>
> Since $(a_n)$ is arithmetic, there exists a common difference $d$ such that $a_n = a_1 + (n-1)d$ for all $n$. From the given information, $a_3 = a_1 + 2d$. Substituting the values: $11 = 5 + 2d$, which gives us $2d = 6$, so $d = 3$. Now we can find $a_5 = a_1 + 4d = 5 + 4(3) = 5 + 12 = 17$.

**PROOFFLOW DAG:** PROOFFLOW DAG generates step-by-step proof faithfully following the logical transition of the natural language proof (Fig. 9).

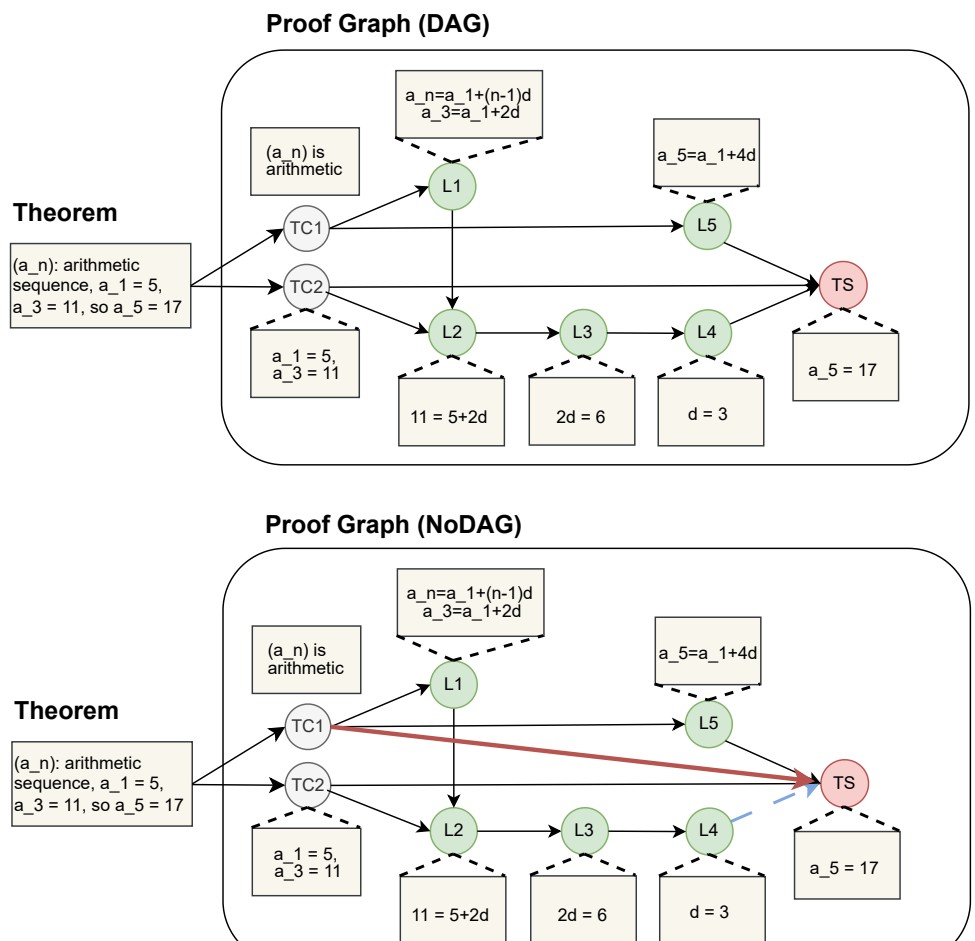

Figure 9: Comparison of proof structures generated by PROOFFLOW DAG (top) and PROOFFLOW noDAG (bottom) with respect to the original natural language proof. The red arrow indicates a dependency in PROOFFLOW noDAG not in the natural language proof. The blue dashed arrow indicates a dependence in the natural language proof but severed in PROOFFLOW noDAG. The one by PROOFFLOW DAG faithfully follows the dependency of the original proof. However, the one by PROOFFLOW noDAG proves the final step TS solely by the two given theorem conditions TC1 and TC2 and step L5. This is not only structurally unfaithful to the structure of the natural language proof but also inefficient as the efforts to prove all intermediate steps L1, L2, L3, L4 are squandered.

**PROOFFLOW noDAG:** The Lean code proof by PROOFFLOW noDAG is not only structurally unfaithful but also inefficient. As illustrated in Fig. 9, the final step TS is proven by the tactics stemming from theorem conditions TC1 and TC2 and intermediate step L5 (direct result of TC1) while neglecting all intermediate steps L1, L2, L3, L4. These steps are proven but not utilized for the final goal of the proof. As a consequence, the logical flow of the proof by PROOFFLOW noDAG fails to adhere to the structure of the natural language proof and squanders the resources used to prove L1, L2, L3, L4.

**FULL PROOF:** FULL PROOF failed to generate syntactically correct Lean code to prove this theorem (the first syntax error is function type error).

**STEP PROOF:** STEP PROOF failed to generate syntactically correct Lean code to prove this theorem (the first syntax error is function type error).

## A.6 EXAMPLES OF ERRORS IN THE NATURAL LANGUAGE PROOF

The errors listed in Table 7 were identified using the thinking mode of our PROOFFLOW DAG pipeline at pass@5. These examples were specifically extracted from proof steps that our error detection pipeline (detailed in Section 4.2) flagged as an "NL statement error."

We have decided to keep these issues in the benchmark because they reflect the kinds of common and subtle ambiguities made by humans when writing proofs. For example, using ambiguous informal language like "we get [result x]" is a frequent occurrence in practice. By including these ambiguities, we ensure the benchmark remains a realistic representation of human-written proofs, rather than an overly artificial one.

| Original Natural Language | Error Type | Comment |
|---|---|---|
| $3a^3 - 3a^2b - 3ab^2 + 3b^3 = (a^2 - b^2)(a - b)$ | Incorrect Statement | An outright algebraic error. The left-hand side is exactly three times the right-hand side. |
| Let $n$ be a natural number. If $n = 1, ...$ If $n$ is prime, we are done. If $n$ is composite, then $n = ab$ ... By induction... | Logical Flaw | The proof sketch is intuitively correct but structurally flawed. It conflates the base case and the inductive step of a strong induction proof. |
| ...we get $(r\frac{\sqrt{3}}{2})^2 + (r/2 - 1)^2 = 1$ | Ambiguity | The phrase "we get" misleadingly suggests a general identity, when this is actually a conditional equation that only holds for the specific value $r = 1$. |
| From the condition $\mathrm{Arg}(z) = \pi/6$, we can write $z = r(\frac{\sqrt{3}}{2} + \frac{i}{2})$ for some $r > 0$. | Missing Assumption | The statement is incomplete because it relies on the unstated precondition that $z \neq 0$ for the $\mathrm{Arg}(z)$ function to be well-defined. |
| Hence, $(X \cdot Y)^2 \leq |X|^2|Y|^2$, because if not, then p would have two distinct real zeros... | Incomplete Argument | The reasoning, based on the discriminant of a quadratic, is incomplete as it fails for the case where $Y = 0$, where the polynomial degenerates. |
| Integrating... yields $v(x, y) = 2xy + 3y + g(x)$, where $g(x)$ is a function of $x$. | Incomplete Statement | The statement is true but insufficient. In the context of solving a PDE, it omits the crucial condition that the function of integration $g(x)$ must also be differentiable. |

Table 7: Analysis of Flaws Identified in Natural Language Proof Steps

## A.7 EVALUATION ON THE MINIF2F BENCHMARK

To validate our findings on an established benchmark, we evaluated 50 randomly selected problems from the miniF2F test set (Zheng et al., 2021). We focused on PROOFFLOWBENCH as our primary benchmark because it represents higher complexity for natural language proofs, where proofs average 637 characters, being 37% larger than miniF2F.

The results, presented in Table 8, confirm that the relative performance ordering of the different pipeline configurations is preserved on this established benchmark. Crucially, all accuracy indicators were universally higher on miniF2F compared to PROOFFLOWBENCH, indicating that miniF2F represents an easier evaluation setting for proof autoformalization.

Table 8: Pass@5 performance of different pipelines on the miniF2F test set (sample of 50 problems).

| Pipeline | Think | Step-Level | | Proof-Level | | | |
|---|---|---|---|---|---|---|---|
| | | Form. Acc. | Tactic Acc. | Proof Score | Correct Syntax | Time (min) | Output (k) |
| PROOFFLOW DAG | No | 0.96 | 0.53 | 0.51 | 0.12 | 6.7 | 19.9 |
| | Yes | 0.99 | 0.88 | 0.63 | 0.58 | 11.9 | 58.4 |
| PROOFFLOW noDAG | No | 0.94 | 0.46 | 0.46 | 0.08 | 9.7 | 23.6 |
| | Yes | 0.98 | 0.83 | 0.53 | 0.50 | 11.8 | 57.9 |
| STEPPROOF | No | / | 0.11 | 0.09 | 0.02 | 0.2 | 1.7 |
| | Yes | / | 0.36 | 0.01 | 0.17 | 6.7 | 22.5 |
| FULLPROOF | No | / | / | 0.12 | 0.14 | 0.8 | 10.7 |
| | Yes | / | / | 0.38 | 0.86 | 2.0 | 7.2 |

## A.8 COMPUTATIONAL EFFICIENCY

We provide comprehensive diagnostics of the pipeline by examining each subcomponent in detail. Table 9 presents key metrics across different pipeline configurations.

Our analysis reveals that the current pipeline bottleneck, in terms of both time and token efficiency, lies primarily in the Tactic Completer stage. The Graph Builder demonstrates high reliability, succeeding on the first attempt in most cases (1.01–1.08 tries per problem). In contrast, the Formalizer typically requires 1–2 attempts depending on the thinking mode, while the Tactic Completer demands significantly more iterations, averaging 2–3 attempts in thinking mode and 3–4 attempts in non-thinking mode per node to generate valid Lean code. The temporal distribution further emphasizes this bottleneck: the Tactic Completer consumes 81–89% of the total execution time across all configurations, with the Graph Builder and Formalizer accounting for substantially smaller fractions.

Regarding token generation, we observe a difference between thinking and non-thinking modes. In thinking mode, the Tactic Completer dominates token usage (81–84%), primarily due to Goedel models generating substantial informal reasoning and extensive thinking before producing formal proofs. Conversely, in non-thinking mode, token distribution is more balanced, with both the Graph Builder and Tactic Completer each accounting for approximately 40% of tokens. This difference stems from the reasoning-heavy output of Gemini-2.5-Pro in the Graph Builder stage, contrasting with DeepSeek-Prover-V2, which was instructed to minimize preliminary reasoning.

Table 9: Performance breakdown of ProofFlow pipeline components across different configurations. Metrics include average number of attempts per component, time distribution percentages, and token usage distribution.

| Metric | ProofFlow DAG | | ProofFlow noDAG | |
|---|---|---|---|---|
| **Think Mode** | No | Yes | No | Yes |
| *Average Attempts* | | | | |
| Graph Builder (per problem) | 1.08 | 1.08 | 1.01 | 1.01 |
| Formalizer (per node) | 2.20 | 1.39 | 1.94 | 1.37 |
| Tactic Completer (per node) | 4.04 | 2.39 | 3.69 | 2.67 |
| *Time Distribution (%)* | | | | |
| Graph Builder | 13.8 | 3.9 | 10.1 | 3.3 |
| Formalizer | 4.9 | 9.8 | 4.7 | 7.9 |
| Tactic Completer | 81.2 | 86.3 | 85.3 | 88.8 |
| **Total Time (mins)** | **8.8** | **31.8** | **12.3** | **32.0** |
| *Token Distribution (%)* | | | | |
| Graph Builder | 44.5 | 10.0 | 36.7 | 8.3 |
| Formalizer | 14.1 | 9.0 | 19.0 | 7.5 |
| Tactic Completer | 41.4 | 81.0 | 44.3 | 84.2 |
| **Total Tokens (k)** | **22.4** | **94.2** | **25.8** | **98.5** |

