# OpenReview forum: "ProofFlow: A Dependency Graph Approach to Faithful Proof Autoformalization"
_ICLR.cc/2026/Conference — ICLR 2026 Poster_

### Official Review · Reviewer_hcof · 2025-10-27

**Soundness:** 3
**Presentation:** 3
**Contribution:** 3
**Rating:** 8
**Confidence:** 3

**Summary:**

This paper focuses on the task of proof autoformalization, distinguishing it from automated theorem proving. The key difference is:

Automated Theorem Proving: A prover constructs a verifiable proof from a given formal statement.

Proof Autoformalization: The model is given both the natural language statement and its proof, and must translate this proof into a formal language while preserving its structure.

The paper introduces ProofFlow, a pipeline that utilizes a DAG (Directed Acyclic Graph) to ensure the final translated proof accurately preserves the logical dependencies between the original proof steps. This pipeline was evaluated on a new benchmark, where it demonstrated good performance.

**Strengths:**

Regarding originality, this paper tackles the under-explored problem of proof autoformalization, specifically focusing on the requirement that the translated proof preserve its structure. This aspect is seldom noticed in previous ATP papers. Moreover, their proposal to use a DAG to address this task is of great novelty.

The paper also provides comprehensive explanations and evaluations of its idea and results. The overall quality and clarity are good, and the evaluations demonstrate that their pipeline achieves good performance.

**Weaknesses:**

No significant weaknesses.

**Questions:**

Do you think it is possible to train a model that can faithfully translate proofs into lean (not relying on your pipeline)? Is it possible to use your pipeline to curate training data for the training of proof formalizer?

---

> ### Author Response · Authors · 2025-11-20
> **Answering question**
>
> Yes, our three-stage pipeline which leverages three different models (one to build the Proof DAG, another to formalize each statement into Lean, and a third to complete the Lean tactics) can be used to curate training data for a single end-to-end proof formalization model.
> Our error detection pipeline (Figure 5) allows us to identify and filter out low-quality autoformalizations, leaving us with higher-quality training data. This opens up several promising research directions:
> - **Supervised Fine-Tuning (SFT):** The three pipeline stages can be embedded as a long chain-of-thought reasoning process, providing training data for a unified model.
> - **Reinforcement Learning:** We can extract error cases from the pipeline to generate negative examples for reinforcement learning techniques like Direct Preference Optimization, where the model learns to prefer correct formalizations over incorrect ones.
> - **Optimized Architecture:** For better efficiency, a two-stage approach may be more effective: A single LLM call performs both DAG construction and formalization for all proof steps. Once formalizations are complete, tactic completion can be parallelized across all steps (since each step's tactic completion is independent once the formalizations are done)
>
> This remains an active area of research. Experiments are needed to determine the optimal approach and whether a fully unified model, a two-stage architecture, or some hybrid configuration yields the best balance of formalization quality, inference speed, and training efficiency. The curated data that can be obtained from our pipeline provides a strong foundation for exploring these different training strategies.
>
> **Paper modification:** We added a small paragraph in the Discussion section, mentioning curating training data as future work.

---

### Official Review · Reviewer_VRSK · 2025-10-31

**Soundness:** 2
**Presentation:** 3
**Contribution:** 3
**Rating:** 6
**Confidence:** 4

**Summary:**

This paper introduces PROOFFLOW, a three-stage pipeline for the proof autoformalization. The pipeline first constructs a directed acyclic graph (DAG) to represent the logical structure of the NL proof (Graph Builder), then translates each node into Lean 4 code (Formalizer), and finally generates tactics to complete the formal proof (Tactic Completer). To support this research, the authors also present two new contributions: PROOFFLOWBENCH, a benchmark of 184 undergraduate-level problems, and PROOFSCORE, a composite metric designed to evaluate the syntactic, semantic, and structural fidelity of the formalized output. The experimental results demonstrate that PROOFFLOW achieves a PROOFSCORE of 0.545, significantly outperforming the baseline methods considered.

**Strengths:**

- Systematic Pipeline Design: The paper presents a well-structured methodology. The PROOFFLOW pipeline thoughtfully deconstructs the complex task of autoformalization into manageable stages. The work is commendably thorough, extending beyond the pipeline itself to include the development of a new benchmark and a tailored evaluation metric.

- Novel "Lemma Approach": A key contribution is the clear conceptual distinction between the conventional low-level "Tactic Approach" and the paper's high-level "Lemma Approach." By demonstrating the superiority of preserving the proof's structure through intermediate lemmas, the paper offers valuable insights that could guide future research in faithful autoformalization.

- Integrated Error Analysis: The inclusion of an error detection system is a significant strength. This mechanism goes beyond simple pass/fail metrics by attempting to diagnose the source of failure, attributing it to the formalizer, the tactic completer, or a potential flaw in the original NL proof. This diagnostic capability is a valuable feature for practical applications.

**Weaknesses:**

- Limited Evaluation Scope: The empirical validation is confined to the newly introduced PROOFFLOWBENCH. The omission of established benchmarks such as miniF2F and ProofNet, which also contain problems with NL proofs, makes it difficult to contextualize the performance of PROOFFLOW within the broader landscape of autoformalization research. A more robust evaluation would compare the proposed method against baselines on these widely recognized datasets.

- Potential Metric Subjectivity and Lack of Validation: The proposed PROOFSCORE metric relies on an "LLM-as-a-judge" to assess semantic faithfulness, introducing a significant risk of subjectivity and unreliability. The evaluation is contingent on the specific LLM employed, and the paper provides no validation for this judge. The absence of an analysis measuring inter-rater reliability (i.e., consistency across different LLMs or against human experts) weakens confidence in the reported scores and the conclusions drawn from them.

- Lack of Transparency and Potential for Data Contamination: The appendix notes that ground-truth dependency graphs for PROOFFLOWBENCH were generated by LLMs before human verification. The paper fails to specify whether the models and prompts used for this data generation are distinct from those used in the pipeline's Graph Builder stage. If they are not, this constitutes a form of data contamination, as the model would be evaluated on a task that closely mirrors its own data generation process. Furthermore, the complete omission of the prompts used for both the pipeline and the metric evaluation is a critical lapse in transparency that hinders the reproducibility of this work.

**Questions:**

In the "Structural Fidelity" evaluation (line 297), the assessment appears to be based on a per-node check of dependencies. Have the authors considered a more holistic assessment of the entire proof structure, for instance, by employing graph structural similarity metrics?

In the experimental setup (lines 399-404), the paper defines "thinking" and "non-thinking" modes with different model configurations for the Formalizer and Tactic Completer stages. Could the authors elaborate on the rationale for selecting these specific model combinations? What hypotheses about model capabilities motivated these distinct configurations? From my understanding, I think it's just about using Gemini-2.5-Pro and Gemini-2.5-Flash in Graph Builder, while the model selection for Formalizer and Tactic Completer should be the same.

---

> ### Author Response · Authors · 2025-11-20
> **Fixing weaknesses and answering questions**
>
> **[R3 - W1]**
>
> We agree that evaluation on established benchmarks strengthens our validation. We focused on our benchmark because it represents higher complexity for natural language proofs, where proofs average 637 characters—37% larger than miniF2F and 80% larger than ProofNet test sets.
>
> We evaluated 50 randomly selected miniF2F test problems. Our findings confirm that conclusions regarding relative performance of different pipeline configurations hold on this established benchmark. Crucially, we observe that all accuracy indicators were universally higher on miniF2F compared to PROOFFLOWBENCH, indicating that miniF2F is an easier benchmark.
>
> | Pipeline | Think | Form Acc. | Tactic Acc. | Proof Score | Proof-Level Correct Syntax | Time (min) | Output (k) |
> |:---|:---:|:---:|:---:|:---:|:---:|:---:|:---:|
> | ProofFlowDAG | no | 0.96 | 0.53 | 0.51 | 0.12 | 6.7 | 19.9 |
> | | yes | 0.99 | 0.88 | 0.63 | 0.58 | 11.9 | 58.4 |
> | ProofFlownoDAG | no | 0.94 | 0.46 | 0.46 | 0.08 | 9.7 | 23.6 |
> | | yes | 0.98 | 0.83 | 0.53 | 0.50 | 11.8 | 57.9 |
> | StepProof | no | - | 0.11 | 0.09 | 0.02 | 0.2 | 1.7 |
> | | yes | - | 0.36 | 0.01 | 0.17 | 6.7 | 22.5 |
> | FullProof | no | - | - | 0.12 | 0.14 | 0.8 | 10.7 |
> | | yes | - | - | 0.38 | 0.86 | 2.0 | 7.2 |
>
>
> Table: Pass@5 performance of different pipelines on the miniF2F test set (sample of 50).
>
> Due to limited time during the review period, we were only able to complete the evaluation of this 50-problem subset. We commit to executing the full evaluation on the complete miniF2F test dataset and include it on the paper, pending acceptance.
>
> **[R3 - W2]**
>
> LLM-based judgment isn't error-free, but [2] demonstrated LLM's reliability for semantic faithfulness. As Claude is not involved in our pipeline modules (Graph builder, Formalizer, Tactic Completer), we use it for a more objective ProofScore evaluation. We validated the reliability of ProofScore by comparing existing results with Gemini-2.5-Flash and DeepSeek-V3 variants (Appendix A.2.3), showing consistency across LLMs.
>
>
> **[R3 - W3]**
>
> Using the same model/prompts is not data contamination because:
>
>  - Gemini-2.5-Pro is closed-source; the initial API calls to help build the ground truth DAG construction are independent of subsequent pipeline testing API calls.
> - We did not use the ground truth DAGs in the paper.
> - A separate model (Claude-Sonnet-4) was used for DAG evaluation (ProofScore's structural fidelity component, Section 4.1 end).
>
> All prompts, code, and replication instructions are on the ProofFlow GitHub, as clarified in Section 3.1 (revised paper).
>
> **[R3 - Q1]**
>
> We initially considered using graph structural similarity metrics to compare the generated DAG against a ground truth DAG. However, we identified several fundamental challenges with this approach:
> - Non-uniqueness of Proof DAGs: As discussed in Section 4.1, there is no single "correct" way to decompose a natural language proof into a DAG, and it is too cumbersome to manually collect all valid proof DAGs.
> - Scalability Beyond the Benchmark: Outside our curated benchmark, ground truth DAGs are not available. Thus, we need an automatic evaluation approach.
> - Interpretability: The per-node approach we use provides actionable feedback about exactly which logical dependencies are violated, whereas a single graph similarity score would be less interpretable for debugging and improvement.
>
>
> **[R3 - Q2]**
>
> We always utilized Gemini-2.5-Pro for the Graph-building stage. The rationale for picking two modes (thinking and non-thinking) with different model choices for the remaining components was to explore the tradeoff between speed and accuracy.
>
> ProofFlow "Thinking" Mode (Prioritizing Accuracy):
> - Graph Builder: We utilized Gemini-2.5-Pro. Our preliminary study of 20 proofs showed it achieved near-perfect structure extraction, significantly outperforming other candidates. We retained this model in the "non-thinking" mode, since we could not find any suitable alternative.
> - Formalizer and Tactic Completer: We selected Goedel-Formalizer-V2 and Goedel-Prover-V2. As indicated in [1], these represent the current state-of-the-art.
>
> ProofFlow "Non-Thinking" Mode (Prioritizing Speed): Here, our objective was to keep total inference time within a reasonable bound (ideally under 10 minutes):
> - Formalizer: We utilized Gemini-2.5-Flash. Our experiments showed it achieves a 75% syntactic correctness rate with lower generation time than the Goedel-Formalizer-V2.
> - Tactic Completer: We employed DeepSeek-Prover-V2. Ranked as the second-best model class in [1], we achieved the necessary speed for this mode, by instructing it to generate answers directly (without extended reasoning).
>
> FullProof and StepProof: We have updated the "Thinking" mode for both FullProof and StepProof to utilize Goedel-Prover-V2 (see response to Reviewer 2CoE). For the "Non-thinking" mode we still use the faster Gemini-2.5-Flash.
>
> -----------------
>
> References
>
> [1] arXiv:2508.03613
>
> [2] arXiv:2506.07047

---

### Official Review · Reviewer_aiML · 2025-10-31

**Soundness:** 2
**Presentation:** 2
**Contribution:** 3
**Rating:** 6
**Confidence:** 3

**Summary:**

This paper presents ProofFlow, a DAG-based framework for faithful proof autoformalization that decomposes natural language proofs into lemma nodes and translates them into Lean 4 via LLMs. It introduces ProofScore, a composite metric for syntactic, semantic, and structural quality, and a new dataset (184 undergraduate proofs). Experiments show its pipline outperforms Full-Proof and Step-Proof in accuracy and structural fidelity.

**Strengths:**

1.	This paper introduces an appealing DAG-based, lemma-driven approach that preserves the logical structure of natural language proofs while improving interpretability and consistency.
2.	It proposes ProofScore, a new metric capturing syntactic correctness, semantic faithfulness, and structural fidelity, providing a more rigorous evaluation.
3.	Experiments demonstrate improvements with ProofFlow, achieving a ProofScore of 0.545 compared to 0.123 (Full-Proof) and 0.072 (Step-Proof), validating its effectiveness and generalizability.

**Weaknesses:**

1. The Formalizer stage contributes to 32–47% of failures due to semantic mismatches between natural language and Lean 4 code, suggesting that its efficiency remains limited.
2. Semantic faithfulness is evaluated through subjective LLM judgments without human verification or inter-rater reliability, which may introduce bias and reduce objectivity.
3. Experiments are restricted to undergraduate-level proofs (average 8.4 nodes per proof, covering elementary topics), leaving the method’s generalizability to research-level or large-scale mathematical corpora untested.

**Questions:**

Please refer to the Weakness section.

---

> ### Author Response · Authors · 2025-11-20
> **Addressing weakeness**
>
> **[R2 - W1]**
>
> We acknowledge the reviewer's observation regarding the high error rate at the Formalizer stage. While many errors are syntactic, semantic errors remain significant. This high semantic error rate is a direct consequence of two factors: our conservative semantic scoring criteria, which intentionally prioritized precision over recall, and the current limitations of formalizer LLMs. Crucially, we emphasize that this error rate is not attributable to the core architectural design of the ProofFlow pipeline, such as our DAG and lemma-based approach. We anticipate that as base formalizer LLMs improve, the bottleneck caused by these semantic errors will diminish.
>
> Strictness of Semantic Alignment: To quantify this, we conducted a manual audit of 200 randomly sampled formalizer errors. A significant portion of the flagged "mismatches" are actually mathematically valid formalizations that were rejected solely due to structural or notational divergences from the natural language. Our system prioritizes "faithfulness to the natural language structure" over "mathematical equivalence," leading to the rejection of valid code in cases such as:
> - Domain Modeling: A prompt might define a sequence index $n$ as "an integer where $n \ge 1$," whereas the model optimizes the formalization by defining $n$ as a "natural number" (Nat).
> - Explicating Implicit Assumptions: Mathematical text often leaves constraints implicit (e.g., that a denominator is non-zero or a set is non-empty). When LLM models correctly identify and add these necessary hypotheses to make the code valid in Lean, the semantic checker often flags them as "hallucinated constraints" because they were not present in the text.
>
>
> **[R2 - W2]**
>
> As we aim to develop an automated workflow, involving human verifications in the pipeline is infeasible.
> While LLM-powered evaluations may not be completely error-free, evidence [3] has suggested the reliability of this scoring method in measuring semantic faithfulness with a proper step-by-step guided approach. To further validate the reliability of ProofScore and the faithfulness score $f_i$, we evaluate the scores it produces with ground-truth results obtained by human experts. The details are in Appendix A.2.3 (revised paper).
>
> **[R2 - W3]**
>
> We thank the reviewer for raising this important issue. While we acknowledge that our experiments focus on undergraduate-level proofs, we believe this scope is appropriate given the current state of the field. The only available benchmark we found that contained research-level natural language proofs was FrontierMath [2], a private benchmark with 12 problems publicly available. As expected, a performance drop was observed compared with ProofFlowBench.
>
> | Pipeline | Form. Acc. | Tactic Acc. | Proof Score | Proof-level Correct Syntax | Time (min) | Output (k) |
> |:---|:---:|:---:|:---:|:---:|:---:|:---:|
> | ProofFlowBench | 0.751 | 0.358 | 0.355 | 0.027 | 8.8 | 22.4 |
> | FrontierMath (10 samples) | 0.432 | 0.192 | 0.099 | 0 | 16.5 | 47.6 |
>
> Table: Performance of ProofFlow DAG (non-thinking) on ProofFlowBench compared with FrontierMath benchmark.
>
> This steep decline is characteristic of the field. [1] showed that even SOTA formalization models drop from 62.7% accuracy on undergraduate problems to 0% on PhD and research level problems, primarily due to hallucinations and the problems being outside the scope of the MathLib library.
>
> We have analyzed our pipeline’s components to identify their generalizability:
> - Graph Builder (not a limitation): We consider this the most generalizable component of our pipeline. This is because the Graph Builder targets the structural semantics of the proof rather than the mathematical content itself. The linguistic markers used to denote flow and dependency (e.g., "therefore," "by Lemma 3," "it follows that") usually remain consistent whether the text is an undergraduate exercise or a PhD-level problem.
> - Formalizer and Tactic Completer (limitation, future work): The limitation arises when the individual lemmas involve research-level concepts; in such cases, performance is bottlenecked by the underlying formalizer and tactic-completer LLMs [1], not by the pipeline architecture. Part of our future work is to curate training data to improve the accuracy of these LLMs in more difficult problems (see comment to Reviewer hcof).
> - Lean and MathLib (limitation, outside the scope of our project): MathLib  currently lacks coverage in specific research domains and research-level formalization often necessitates user-defined definitions that are absent from the standard library.
>
> In summary, our pipeline which relies on the semantic decomposition of proofs, is designed to generalize to more difficult problems as underlying LLMs and MathLib library mature. We made a comment related to this issue in the Discussion Section (revised paper).
>
> --------------------
>
> References:
>
> [1] arXiv:2511.02872
>
> [2] arXiv: 2411.04872
>
> [3] arXiv:2506.07047

---

### Official Review · Reviewer_2CoE · 2025-11-01

**Soundness:** 2
**Presentation:** 3
**Contribution:** 4
**Rating:** 4
**Confidence:** 4

**Summary:**

The idea of this paper is to autoformalize proofs faithfully, meaning the semantic structure of the proof is maintained rather than just the correctness. The authors provide a new formalization pipeline called ProofFlow, in which they decompose the proof into a DAG where each node is a proof step, and edges indicate dependency. They also introduce scoring system to evaluate faithfulness (ProofScore), and a benchmark dataset of undergrad problems. They also conduct experiments to support their claims.

**Strengths:**

The paper introduces a novel approach to proof autoformalization which enforces structural fidelity. The authors make a good case for why this is an important problem, and why their proposed method would do so (if effective). Furthermore, the provided dataset and software seems like a useful contribution to the autoformalization/ATP community. Additionally, their pipeline and scoring metric both seem like reasonable and inuitive approaches to the problems.

**Weaknesses:**

I think while your method has merit, the comparisons to previous methods is not exactly fair. You are using a un-finetuned Gemini to formalize and prove proofs/steps for the previous methods, whereas ProofFlow uses much stronger, (nearly) state-of-the-art autoformalization/ATP models in the Goedel models. While this might be tricky for Full Proof (since Goedel models are not trained for proof-autoformalization), I think a more fair comparison for Step Proof would be

1. Break down the proof into steps using the same model as you used for GraphBuilder
2. Prove each step using Goedel Formalizer/Prover

In this way you're normalizing for the strength of the model to ensure you're comparing the method. Without this (or another way to ensure the model strength doesn't affect performance), the performance gain is difficult to believe.

Secondly, ProofFlow requires significantly more computation than previous methods, which is a limitation of the work. I believe it would benefit from some analysis on computationally demanding elements of the pipeline (e.g., how many iterations is usually required to make a valid DAG in the GraphBuilder step?). This would offer some avenues for improvement to mitigate this limitation.

**Questions:**

L192: This approach assumes each proof step depends on all preceding steps, a simplification that can lead to unintended consequences.

Figure 2: L3 typo, should be ^2 I think. Also, it seems to me that L3 is entirely redundant. Why not simply use L2 and L5 to get to L6? Is this a fault of the system(s) used? Possibly showing the full informal proof would be helpful.

Figure 5: Could be the case that multiple errors occured, i.e. a NL statement error could be hidden behind a tactic completer error. Does this happen?

* I'm confused on the ProofScore metric. When you say syntactic correctness, my understanding is that you are just concerned about whether a statement passes the Lean compiler, regardless of the "fidelity" to the original informal statement. For example, is "theorem abc : True := by trivial" considered syntactically correct?

* I'm also not sure how to interpret the ProofScore metric. I think showing some examples of proofs with various proof scores would be useful to help readers understand how the numbers vary.

---

> ### Author Response · Authors · 2025-11-20
> **Fixing weaknesses and answering questions**
>
> **[R1- W1] Comparison study**
>
> We agree with the reviewer that utilizing the specialized Goedel models within the FullProof and StepProof methods is essential for a fair comparison with the pipeline we propose. We note that for StepProof, the initial proof-decomposition phase employs the same model and prompt as ProofFlow; this detail is now clarified in Section 6.1 of the updated paper. Consequently, we have updated Table 1 and Table 3 with new results, specifically by utilizing Goedel-Prover-V2-32B for the "thinking" mode in both the FullProof and StepProof pipelines. This change yielded a significant increase in proof-level syntactic accuracy for the FullProof (thinking) method, which is now 0.571. However, a substantial performance disparity remains in the ProofScore evaluation: the FullProof (thinking) pipeline achieves 0.279, while the ProofFlow DAG pipeline (thinking) achieves 0.545. A more detailed justification for the selection of these model configurations is provided in our response to Reviewer VRSK.
>
>
>
> **[R1 - W2] Computational Effort**
>
> We have enhanced our analysis of the pipeline's computational efficiency ) by adding detailed diagnostics in the new Appendix Section A.7 (and Table 8), and by including a new paragraph on computational efficiency in Section 6.2 of the main text.
>
> Our analysis reveals that the current pipeline bottleneck, in terms of both time and token efficiency, lies primarily in the Tactic Completer stage. The Graph Builder demonstrates high reliability, succeeding on the first attempt in most cases (1.01--1.08 tries per problem). In contrast, the Formalizer typically requires 1--2 attempts depending on the thinking mode, while the Tactic Completer demands significantly more iterations, averaging 2--3 attempts in thinking mode and 3--4 attempts in non-thinking mode per node to generate valid Lean code. The temporal distribution further emphasizes this bottleneck: the Tactic Completer consumes 81--89\% of the total execution time across all configurations, with the Graph Builder and Formalizer accounting for substantially smaller fractions.
>
> A straightforward solution to decrease wall time, enabled by our pipeline design, is to parallelize the Tactic Completer calls across different nodes.  With our DAG approach, both the formalizer and tactic completer stages can be parallelized. First, we can formalize proof nodes in different branches across separate threads; also, once a node has been formalized, the corresponding nodes can be tactic-completed completely in parallel. In principle, our pipeline enables this extensive parallelization. However, we did not implement it due to limited computational resources— our GPU resources only allowed us to use the formalizer and tactic completer one item at a time. For evaluation fairness, we also did not parallelize the non-thinking mode, even though we could have parallelized the API calls. We added a new paragraph in Section 6.2 with these conclusions.
>
>
>
> **[R1 - Q1] L192**
>
>  We have rephrased the paragraph containing this line for clarity.
>
>
>
> **[R1 - Q2] Figure 2 typo**
>
> Thank you for pointing out the mistake. We have corrected the figure. The natural proof structurally follows the DAG in which L3 and L5 are needed to prove L6. As explained in the caption, without enforcing DAG, a prover may reuse L1 to prove L3 while skipping L2.
>
>
> **[R1 - Q3] Figure 5**
>
> Yes, multiple errors can occur. For instance, the initial NL statement may be incorrect, and the formalizer may unfaithfully translate it into Lean. In this case, the pipeline will only detect a formalization error.
>
> We emphasize that the error detection pipeline depicted in Figure 5 is not designed for grading NL statements. It is a diagnostic tool to understand and improve our ProofFlow autoformalization pipeline and to detect the current weakest links. Because an unfaithful translation makes the following pipeline steps useless, our system is programmed to stop and report the first error it finds, following this order:
>
> 1. Formalizer Error
> 2. Tactic Completer Error (inability to prove/disprove)
> 3. NL Statement Error (if the Tactic Completer disproves the statement)
>
> Future work could focus on building dedicated tools for NL error detection, as the goal of the error detection pipeline is to diagnose ProofFlow pipeline weaknesses.
>
>
> **[R1 - Q3-1] ProofScore metric**
>
> Yes, we consider "theorem abc : True := by trivial" to be syntactically correct, even though the natural language statement it attempts to formalize may have a completely different meaning. This is precisely why the ProofScore metric incorporates multiple factors beyond syntactic correctness, and we multiply $c_i$ (syntactic correctness) with $f_i$ (semantic faithfullness) of node $i$.
>
>
> **[R1 - Q3-2] ProofScore Understanding**
>
> To facilitate a better understanding, we have provided several pairs of NL proof - Lean code with varying Proof Scores in the revised paper. Please refer to Appendix A.2.4.

---

### Author Response · Authors · 2025-11-21
**Response to Reviewer Feedback and Manuscript Revisions**

We sincerely thank the reviewers for their insightful comments and constructive feedback.

We confirm that we have addressed all individual reviewer questions directly within their respective submission boxes. In addition, we have updated our manuscript  to incorporate the suggested changes and clarifications.

All section references provided in our individual responses correspond to this newly revised version of the manuscript. Due to the short character limit imposed on the individual reviewer response boxes, we were constrained to provide only a very succinct reply in some cases.

We are keen to provide any further details or clarification that may be needed to fully address any remaining concerns.

---------------------------------
**Summary of Main Manuscript Modifications**

The key modifications and new analyses incorporated into the revised manuscript are summarized below:

- Enhanced Comparative Study (R1-W1): We updated our comparison against the FullProof and StepProof methods by utilizing the specialized Goedel-Prover-V2-32B model in their "thinking" modes for a fair comparison. Table 1 and Table 3 reflect these new results. A significant performance gap remains in the ProofScore evaluation: FullProof (thinking) achieved 0.279 while our ProofFlow DAG pipeline (thinking) achieved 0.545.

- Detailed Computational Efficiency Analysis (R1-W2): We added a new Appendix Section A.7 (and Table 8) and a paragraph in Section 6.2 with detailed computational diagnostics. The analysis confirms the Tactic Completer is the primary bottleneck, consuming 81–89% of the total execution time. We highlight in Section 6.2 that the DAG structure inherently enables extensive parallelization of the Formalizer and Tactic Completer stages.

- Error Analysis and Semantic Strictness (R2-W1, R2-W2): We clarified that the high error rate at the Formalizer stage is due to strict semantic scoring and current LLM limitations, not the core ProofFlow architecture. A manual audit confirmed many rejected "mismatches" are mathematically valid formalizations flagged due to structural/notational divergences. We further validated the reliability of our LLM-powered metric, ProofScore, in Appendix A.2.3.

- Generalizability and Benchmark Expansion (R2-W3, R3-W1):

    - miniF2F Evaluation: We performed a Pass@5 evaluation on a 50-problem subset of the miniF2F test set. This evaluation confirmed that the conclusions regarding the relative performance of different pipeline configurations hold on this established benchmark. We noted that accuracy was universally higher on miniF2F compared to PROOFFLOWBENCH, indicating miniF2F is an easier benchmark.

    -  Research Level evaluation: We included an evaluation on a small sample of the research-level FrontierMath benchmark. We discussed that Graph Builder is highly generalizable, but the Formalizer/Tactic Completer performance depends on LLM and MathLib maturation.

- Data Contamination Clarification (R3-W3): We clarified that no data contamination risk exists because the ground truth DAGs were not used in this project, and a separate model was used for ProofScore evaluation.

----------

We look forward to further discussion and incorporating your valuable feedback to improve our paper.

Best regards,

The Authors

---

### Author Response · Authors · 2025-11-27
**Rebuttal Phase Closing Soon**

Dear Reviewers,

We hope this message finds you well.

This is a gentle follow-up regarding your reviews for our submission. We have incorporated all the suggested revisions, clarified ambiguities, and provided new analyses in the updated manuscript and individual responses.

As the rebuttal deadline is approaching in six days, we are writing to politely encourage you to let us know if our replies have successfully addressed your main concerns. Your final assessment during this rebuttal window is vital for us to address any outstanding issues or concerns that might improve our paper.

Thank you again for your time and expertise.


Best regards,

The Authors

---

### Meta-Review · Area_Chair_7Uy5 · 2026-01-06

**Summary:**

Reviewer 2CoE.
 - Unfair baseline comparisons. Prior methods were run with a weaker/unfine-tuned model (Gemini) while ProofFlow used stronger Goedel models, so gains may reflect model strength rather than method. They asked to normalize baselines with the same Goedel components.
 - High compute cost / missing efficiency analysis. ProofFlow appears much more expensive; they wanted breakdowns such as iteration counts (e.g., how many tries to build a valid DAG) and identification of the bottlenecks.
 - Clarity issues about ProofScore + error taxonomy. Questions about what “syntactic correctness” means (e.g., whether theorem : True := by trivial counts), how to interpret ProofScore without examples, whether multiple errors can be masked, and some text/figure clarifications (L192, Fig. 2 typo/redundancy).

Reviewer aiML  (first time reviewer)
 - Formalizer bottleneck. The Formalizer accounts for a large fraction of failures (claimed 32–47%) due to semantic mismatch, suggesting limited effectiveness at a key stage.
 - Subjective semantic faithfulness evaluation. Reliance on LLM judgments without human verification and inter-rater reliability may bias the semantic-faithfulness component of ProofScore.
 - Generalization limits. Experiments are restricted to undergraduate-level proofs (184 items, about 8.4 nodes per proof), leaving research-level scaling and broader applicability untested.

Reviewer VRSK  (first time reviewer)
 - Evaluation scope / missing standard benchmarks. Results are only on the new ProofFlowBench; they wanted comparisons on established benchmarks (explicitly miniF2F, ProofNet) to contextualize performance.
 - ProofScore reliability not validated: Concern that LLM-as-judge introduces subjectivity; asked for validation such as agreement across different judges or vs human experts (inter-rater reliability).
 - Transparency + possible data contamination. Ground-truth DAGs were initially generated by LLMs before human checking; the paper did not clearly separate those models/prompts from the tested Graph Builder, raising contamination risk. Also criticized omission of prompts as hurting reproducibility.
 - Design rationale questions. Asked about using holistic graph similarity metrics for structural fidelity, and about the rationale for “thinking vs non-thinking” model configurations.

This VRSK review is meaningful, although it is likely an LLM-generated review. The principal chunk of the authors last response (Summary of Main Manuscrpt Modifications) on November 21 is likely also LLM-generated.

Reviewer hcof - low quality review

**Reviewer Concerns:**

Reviewer 2CoE

Addressed
- Baseline fairness. Authors reran and updated baselines to use stronger Goedel components (esp. for “thinking” mode) and clarified that StepProof’s decomposition uses the same setup as GraphBuilder (per rebuttal).
 - Compute cost. Added component-level diagnostics (attempt counts + time/token shares), identifying the Tactic Completer as the bottleneck and showing Graph Builder typically succeeds in  one try per problem.
 - ProofScore clarity. Explicitly states that syntactically-valid but semantically-vacuous Lean (e.g., proving True) still counts as “syntactic correctness” and that faithfulness is handled by other factors, adds worked examples of scores.

Still outstanding
 - Even with improved fairness, one could still argue comparisons depend on budgeting/attempt policies and other pipeline details (not fully method-isolated).
- ProofScore validation is improved but still mostly LLM-mediated (see aiML/VRSK points), 2CoE’s concerns are largely mitigated, but not fully eliminated.

Reviewer aiML

Addressed
 - Metric subjectivity / lack of validation. Appendix adds a human-expert comparison on 200 proof steps, reporting F1 for “PASS/FAIL semantic equivalence,” and shows similar behavior across multiple judge-LLMs.
 - ProofScore interpretability. Adds concrete NL - Lean examples with different ProofScore outcomes.

Still outstanding
 - Formalizer bottleneck remains (authors argue it’s partly due to strict faithfulness criteria and current model limits), but the pipeline still fails often at that stage in practice.
 - Generalization is still weak, evidence on truly research-level corpora remains limited; rebuttal acknowledges performance drop and attributes it to underlying models/Mathlib.

Reviewer VRSK

Addressed
 - Scope and standard benchmarks. Authors add a miniF2F subset evaluation (50 problems) and claim trends persist (per rebuttal).
 - ProofScore reliability. Appendix provides cross-LLM consistency + human-expert check for semantic faithfulness judgments.
 - Transparency (prompts/code). The paper states that the repo includes all LLM prompts and provides replication instructions.

Still outstanding
 - Benchmark coverage is still partial (miniF2F only a subset, ProofNet not shown, “full eval after acceptance” is not evidence during review).
 - The data-contamination worry is only partly neutralized, the paper text does not clearly spell out  a fully disjoint “data-generation model/prompt” vs “tested GraphBuilder model/prompt” protocol, even if the repo may contain that detail.

**Reviewer Scores:**

The scores would most likely remain unchanged overall, with possibility of small raise by some reviewers, but also a possibility of lowering their scores by other reviewers. The positive reviewer hcof could provide more meaningful remarks during the discussion, which could improve the  small weight of the otherwise low quality hcof review.

---

### Decision · Program_Chairs · 2026-01-26

Accept (Poster)